# Finite-Time Analysis of Federated Temporal Difference Learning with Linear Function Approximation under Environment and Computation Heterogeneity

## Abstract

Temporal difference (TD) learning is a popular method for reinforcement learning (RL). In this paper, we study federated TD learning of multi-agent systems, where each agent collaboratively performs policy evaluation with linear function approximation under heterogeneous computation configurations and environments. We devise a Heterogeneous Federated TD (HFTD) algorithm which iteratively aggregates agents' local stochastic gradients for TD learning. The HFTD algorithm involves two major novel contributions: 1) it aims to find the optimal value function model for the mixture environment where the environment is randomly drawn from the heterogeneous environments, and the procedure relies only on the local gradients arising from each agent's mean squared projected Bellman errors (MSPBEs) for their respective environments; 2) it allows agents to perform different numbers of local iterations for TD learning, thus enhancing the computational capacity of heterogeneous agents. We analyze the finite-time convergence of the HFTD algorithm for the scenarios of I.I.D. sampling and Markovian sampling. By characterizing bounds on the convergence error, we show that the HFTD algorithm can asymptotically converge to the optimal model, which is the first result in existing works on federated RL to our best knowledge. The HFTD algorithm also achieves a sample complexity of $\mathcal{O}\left(\frac{1}{N\varepsilon^2}\right)$ and linear convergence, which match the results of existing TD algorithms.

## 1 Introduction

Reinforcement learning (RL) is a promising machine learning (ML) paradigm and RL has been applied for a wide range of sequential decision-making problems, such as autonomous driving, gaming and dynamic resource allocation (Mnih et al., 2013; Ye et al., 2019; Kiran et al., 2021). One major challenge for RL is that, to achieve a high learning accuracy, an agent needs to learn the optimal policy in an online manner from a large amount of data samples via repeatedly interacting with the environment. Hence, the learning performance is restricted by the computational efficiencies of the agents. For example, when RL is applied to train a walking robot, the walking ability of the robot is limited by the number of samples used by the robot (Ibarz et al., 2021; Rudin et al., 2022).

Along a different avenue, as an emerging ML paradigm, federated learning (FL) carries out model training in a distributed manner (McMahan & Ramage, 2017): Instead of transmitting data from a possibly large number of clients to a central server for training, FL trains a global ML model by aggregating local ML models computed distributedly by clients based on their local data (Li et al., 2020; Karimireddy et al., 2020; Huang et al., 2022). One significant advantage of FL is to preserve the privacy of each client data. Moreover, since only local ML models rather than local data are sent to the server, the communication costs can be greatly reduced. Furthermore, FL can exploit substantial computation capabilities of ubiquitous smart devices (Yang et al., 2020).

A natural idea to deal with the sample-efficiency issue of RL is to share the information among multi-agents, so that agents can accelerate the process of learning the optimal policy based on other's experiences. However, in practical applications, raw RL trajectories often contain sensitive information

(e.g., medical information of patients) and thus sharing them compromises agents' privacy. Moreover, exchanging data samples among agents usually leads to high communication costs. Hence, inspired by FL, *federated reinforcement learning* (FRL) has been proposed as a promising approach which allows agents to collectively learn the global optimal policy without sharing raw dataset (Jin et al., 2022; Fan et al., 2021; Wang et al., 2023; Zeng et al., 2021; Qi et al., 2021).

As federated supervised learning which has been widely studied, FRL meets similar challenges, including data (environment) heterogeneity and system (computation and communication) heterogeneity. Moreover, there are some unique challenges in FRL due to its salient features. In particular, unlike typical supervised learning where data are available offline and data samples are drawn from I.I.D. distributions, each agent in FRL collects data *online* by following its own *Markovian* process decided by its behavior policy and state transitions of the environment.

Compared to the performance guarantees in parallel RL where agents interact with identical environments independently (Khodadadian et al., 2022; Liu & Olshevsky, 2023; Fan et al., 2021), studies on FRL with heterogeneous environments are still very limited due to the challenges mentioned above. For a few recent works on FRL where agents have heterogeneous environments (including federated TD learning (Jin et al., 2022; Wang et al., 2023)), *none* of these works has shown that the convergence of their algorithms can diminish to zero asymptotically (i.e., the convergence error can be made arbitrarily small with appropriate hyperparameters of the algorithm such as the number of rounds and step size). Moreover, the existing works on FRL assume that each agent can only use the *same* number of local iterations in each round of the algorithm. However, in practice the computation capabilities of agents can be highly diverse, and stragglers could significantly slow down the training process. Therefore, it is more efficient for agents to use different local iterations based on their computation capabilities.

To tackle the challenges above, in this paper, we focus on federated TD learning for policy evaluation with linear function approximation, where agents interact with heterogeneous environments modeled as Markov Decision Processes (MDPs), and collaboratively learn the value function model for a given policy. The MDPs have identical state spaces and action spaces, but heterogeneous transition probability kernels and reward functions. We also consider agents with heterogeneous computation capabilities, and they can use different numbers of local iterations in each round of the federated TD learning. Given the *environment* heterogeneity and *computation* heterogeneity of agents, we aim to answer the following fundamental questions: 1) Is it possible to design a federated TD algorithm that *asymptotically converge to the optimal value function model*? 2) If yes, what is the *sample complexity* of this algorithm? .

We highlight the main contributions of this paper as follows.

- We study federated TD learning with linear function approximation, where multiple agents collaboratively perform policy evaluation via TD learning, while interacting with heterogeneous environments and using heterogeneous computation configurations. We devise a *Heterogeneous Federated Temporal Difference* (HFTD) learning algorithm which iteratively aggregates agents' local stochastic gradients for TD learning. Compared to existing works on FRL, the HFTD algorithm involves two major novel elements: 1) The goal of the algorithm is to find the optimal value function model for the environment that is *randomly drawn* from agents' heterogeneous environments. To this end, the algorithm aims to minimize the *average* of agents' *mean squared projected Bellman errors* (MSPBEs) for their respective environments, using stochastic gradients of the MSPBEs; 2) The HFTD algorithm allows agents to perform different numbers of local iterations for TD learning. In particular, each agent's local stochastic gradients in a round are *averaged* by its local iteration number, which is then used for local gradient aggregation by the server.

- For the settings of I.I.D. sampling and Markovian sampling, we analyze the finite-time convergence performance of the HFTD algorithm by characterizing the bounds on the convergence error. Our results show that the HFTD algorithm can asymptotically converge to the *optimal* value function model as the number of rounds go to infinity with appropriate step sizes. To our best knowledge, this is the *first* result in existing works on federated RL with heterogeneous environments that the convergence error can diminish to zero asymptotically. In particular, a key property of the global gradient of the average MSPBE allows us to remove a *non-vanishing bias* in the convergence error, so that only vanishing error terms are left. We also show that the HFTD algorithm achieves a sample complexity of

Table 1: Comparison of settings and results for federated temporal difference learning

| References | Vanishing Convergence Error | Heterogeneous Local Iteration Numbers | Heterogeneous Environments | Markovian Sampling |
|---|---|---|---|---|
| (Liu & Olshevsky (2023)) | √ | × | × | × |
| (Khodadadian et al. (2022)) | √ | × | × | √ |
| (Wang et al. (2023)) | × | × | √ | √ |
| This paper | √ | √ | √ | √ |

$\mathcal{O}\left(\frac{1}{N\varepsilon^2}\right)$ and linear convergence speedup, which match the results of existing TD learning algorithms Bhandari et al. (2018); Khodadadian et al. (2022).

## 2 RELATED WORK

**Temporal Difference Learning.** Most existing works on TD learning have focused on the case of a single agent. For TD learning under I.I.D. sampling, the asymptotic convergence has been well studied in Borkar & Meyn (2000); Borkar (2009), and the non-asymptotic convergence (i.e., finite-time analysis) has been studied in Kamal (2010); Dalal et al. (2018); Thoppe & Borkar (2019). For TD learning under Markovian sampling, the asymptotic convergence has been studied in Tsitsiklis & Van Roy (1997); Tadić (2001), and the non-asymptotic analysis has been studied in Bhandari et al. (2018); Srikant & Ying (2019); Hu & Syed (2019); Xu et al. (2020b).

**Federated Learning.** FL has emerged as a groundbreaking computing paradigm for ML by democratizing the learning process for potentially many clients (McMahan & Ramage, 2017; Bonawitz et al., 2019; Stich, 2019; Li et al., 2020; Wang & Ji, 2022; Zhang et al., 2021b; 2022; Guo et al., 2022; Huang et al., 2022; Karimireddy et al., 2020; Fallah et al., 2020; Cui et al., 2021). Compared to conventional distributed ML, FL has several salient features, including multiple local iterations of clients, and heterogeneous computation capabilities of clients. Prior works on FL predominantly focused on federated supervised learning.

**Federated Reinforcement Learning.** The settings of FRL have significant differences from those of federated supervised learning, due to the salient features of RL, including online data sampling (especially Markovian sampling), and dynamic state transition. There have been a few recent works on FRL (Xie & Song, 2023; Zeng et al., 2021; Fan et al., 2021; Liang et al., 2023). However, most of these works have not considered FRL with heterogeneous environments and heterogeneous computation configurations of agents.

**Distributed Reinforcement Learning.** Distributed reinforcement learning (DRL) considers multiple agents operating in a distributed fashion. As a major class of DRL, *parallel* RL (Fan et al., 2021; Mnih et al., 2016) uses multiple learners to solve a large-scale single-agent RL task (Li & Schuurmans, 2011; Nair et al., 2015), where the learners aim to learn a common policy for different instances of the *same* environment. Another major class of DRL is multi-agent reinforcement learning (MARL), where a group of agents operate in a common environment; all agents' actions influence the global state transition and MARL aims at seeking the optimal policy combining all local policies in a collaborative manner (Zeng et al., 2022; Zhang et al., 2018), or find local optimal policies in a non-collaborative manner (Zhang et al., 2021a). These prior works of DRL are different from FRL, since 1) agents in FRL can interact with diverse environments and collaboratively learn a common policy in different environments; 2) FRL involves some unique features of FL, including multiple local iterations of agents, heterogeneous and time-varying computation capabilities of agents.

Some recent works have studied FRL with *heterogeneous* environments (Jin et al., 2022; Wang et al., 2023). However, *none* of the algorithms in these works have convergence guarantee with vanishing errors. In this paper, we show that the proposed HFTD algorithm can asymptotically converge to the *optimal* value function model, which is the *first* such result in existing works on FRL with heterogeneous environments. Moreover, agents can have *heterogeneous* numbers of local iterations in the HFTD algorithm. We have elaborated the technical differences in the convergence analysis of the HFTD algorithm in Section 4.3.

## 3  PRELIMINARIES

**Policy Evaluation in a Single-Node Setting.** We consider the problem of evaluating a given policy in an MDP defined by the tuple $\mathcal{M} \triangleq \{\mathcal{S}, \mathcal{A}, \mathcal{P}, \mathcal{R}, \gamma\}$, where $\mathcal{S}$ is the set of states, $\mathcal{A}$ is the set of actions, $\mathcal{P}$ is the Markovian transition kernel, $\mathcal{R}$ is a reward function, and $\gamma$ is the discount factor. For a discrete state space $\mathcal{S}$, $\mathcal{P}(s'\,|s\,)$ shows the probability of transitioning form $s$ to state $s'$. The reward function of $s$ is denoted by $\mathcal{R}(s) = \sum_s \mathcal{P}(s'\,|s\,)\mathcal{R}(s, s')$. For a given policy $\mu$, the expected cumulative future rewards can be represented as a function of the initial state $s$:

$$V_\mu(s) = \mathbb{E}\left[\sum_{t=0}^{\infty} \gamma^t R_\mu(s_t)\,|s_0 = s\right],$$

where $\{s_t\}$ is the sequence of states generated by the transition kernel $P_\mu$. The value function satisfies the Bellman equation $T_\mu V_\mu = V_\mu$, where for any $V \in \mathbb{R}^{|S|}$,

$$(T_\mu V)(s) = R_\mu(s) + \gamma \sum_{s'} P_\mu(s, s')V(s'). \tag{1}$$

**TD Learning with Linear Function Approximation.** To mitigate the effect of intractable computation in face of large state spaces in policy evaluation, a common and tractable approach is to utilize linear function approximator for a representation of value functions. Let $\{\Phi_k\}_{k=1}^{d}$ be a set of $d$ linearly independent basis vectors in $R^n$. For the value function approximation in this paper, the true value function $V_\mu$ is approximated as $V_\mu(s) \approx V_\theta(s) = \phi(s)^{\mathrm{T}}\theta$, where $\phi(s) \in R^d$ is a fixed feature vector for state $s$ and $\theta \in \mathbb{R}^d$ is the unknown model to be learned. At time $t$, for an observed tuple $O_t = (s_t, r_t, s_{t+1})$, the TD target can be defined using Bellman operator as (1). Then the loss function is defined as the squared Bellman error (Bhandari et al., 2018). The negative gradient of the Bellman error evaluated at the current parameter $\theta_t$ can be expressed as

$$g_t(\theta_t) = (r_t + \gamma\phi(s_{t+1})^{\mathrm{T}}\theta_t - \phi(s_t)^{\mathrm{T}}\theta_t)\phi(s_t). \tag{2}$$

Then the estimated model at time $t + 1$ can be updated by the gradient descent method (Bhandari et al., 2018) with step size $\alpha \in (0, 1)$ as

$$\theta_{t+1} = \theta_t + \alpha g_t(\theta_t). \tag{3}$$

When state $s_t$ sampled in tuple $O_t$ follows the stationary distribution of the MDP, the expected negative gradient at $\theta$ is

$$\bar{g}_t(\theta) = \sum_{s_t, s_{t+1}} \pi(s_t)P(s_{t+1}\,|s_t\,)(R(s_t, s_{t+1}) + \gamma\phi(s_{t+1})^{\mathrm{T}}\theta - \phi(s_t)^{\mathrm{T}}\theta)\phi(s_t), \tag{4}$$

where $\pi(\cdot)$ is the stationary distribution of the associated Markov chain which is assumed to be irreducible and aperiodic.

Let $D$ denote the diagonal matrix whose elements consist of the entries of $\pi(\cdot)$. In the convergence analysis of TD(0) using (3), Tsitsiklis & Van Roy (1997) has proved the limit point $\theta^*$ is the unique solution to the projected Bellman equation $\Phi\theta = \Pi_D T_\mu \Phi\theta$ with $\bar{g}(\theta^*) = 0$, where $\Pi_D(\cdot)$ is the projection operator defined on the subspace $\{\Phi x\,|x \in R^d\,\}$.

## 4  HETEROGENEOUS FEDERATED TEMPORAL DIFFERENCE LEARNING WITH LINEAR FUNCTION APPROXIMATION

In this section, we first describe the problem statement about the federated policy evaluation in heterogeneous environments where agents collectively seek to find a global model to approximate the value function under a given policy $\mu$. As we discussed above, in the process of policy evaluation for a single agent $i$, the local loss function $F_i$ is usually defined as expected Bellman error squared (Bhandari et al., 2018; Srikant & Ying, 2019). Accordingly, the optimization problem for federated value evaluation can be formulated as

$$\min_{\theta \in \mathbb{R}^d} \left[F(\theta) = \frac{1}{N}\sum_{i=1}^{N} F_i(\theta)\right] \tag{5}$$

where $F_i(\theta) = \mathbb{E}_{O_t \sim \pi_i} \left[ \frac{1}{2}(r_t + \gamma V_\theta(s_{t+1}) - V_\theta(s_t))^2 \right]$ is the local objective function of $i$-th agent, i.e., the expected squared Bellman error with respect to the model parameter $\theta$. Here, $\pi_i$ is the stationary distribution of the state transition Markov chain in $i$-th environment, and $O_i^t = (s_i^t, r_i^t, s_i^{t+1})$ represents a data sample from the environment $i$. We assume that each agent collects samples by interacting with its own environment independently. The MDP of agent $i$ can be expressed by: $\mathcal{M}_i \triangleq \{\mathcal{S}, \mathcal{A}, \mathcal{P}_i, \mathcal{R}_i, \gamma\}$. We assume that all agents have the same state space and action space while the reward functions and transition probability functions may differ across various agents.

Prior works on FRL with heterogeneous environments (Jin et al., 2022; Wang et al., 2023) considered the objective of optimizing the value function model for a *single virtual* environment. This virtual environment is constructed as an MDP $\bar{\mathcal{M}} \triangleq \{\mathcal{S}, \mathcal{A}, \bar{\mathcal{P}}, \bar{\mathcal{R}}, \gamma\}$ by averaging the transition kernels and reward functions of each agent's environment, given by $\bar{\mathcal{P}} = (1/N) \sum_{i=1}^N \mathcal{P}_i$ and $\bar{\mathcal{R}} = (1/N) \sum_{i=1}^N \mathcal{R}_i$. However, such an "averaged" environment may not coincide with any agent's individual environment. Intuitively, from the perspective of an individual agent, the objective function may not encourage more agents to participate in the federation. Motivated by this observation, in this paper, we consider a *mixture* environment defined as the environment *randomly drawn* from agents' heterogeneous environments. Note that this mixture environment is different from the virtual environment in Jin et al. (2022); Wang et al. (2023). Thus our goal is to find the optimal value function model for this mixture environment of agents. This goal is *similar* in spirit to that of federated supervised learning, since the latter aims to find the optimal model that minimizes the average training loss for all data samples of all clients. Towards this goal, the global objective function of our federated TD learning problem is the *average MSPBE* of all agents for their respective environments, as the MSPBE quantifies the error of a value function model for an environment. To minimize the average MSPBE, we devise the HFTD algorithm which updates the value function model via federated TD learning. The algorithm aims to iteratively estimate the gradient of the global objective function (i.e., the average MSPBE), given by

$$\bar{g}(\theta) = \frac{1}{N} \sum_{i=1}^N \bar{g}_i(\theta) \tag{6}$$

where $\bar{g}_i(\theta)$ is the gradient of agent $i$'s local objective function (i.e., the MSPBE for agent $i$'s environment). Note that the condition (6) is substantially different from the estimation in Wang et al. (2023) (6) does not hold in Wang et al. (2023)), and is also a key property that allow us to show that our HFTD algorithm can asymptotically converge to the optimal model $\theta^*$ (rather than to a neighborhood of $\theta^*$ where the radius of the convergence error is determined by some non-vanishing bias error as in Khodadadian et al. (2022), see Section 4.3 for detailed discussions of the technical differences). The optimal value function model $\theta^*$ that minimizes the average MSPBE of agents satisfies $\bar{g}(\theta^*) = 0$. Note that the gradient in TD learning is different from that of the standard gradient descent, as $\bar{g}_i(\theta)$ or $\bar{g}(\theta)$ is not the gradient of any *fixed* objective function. To estimate the gradient $\bar{g}(\theta)$, the HFTD algorithm computes a stochastic gradient $g(\theta_t)$ given by $g(\theta_t) = \frac{1}{N} \sum_{i=1}^N g_i(\theta_t)$, where $g_i(\theta_t)$ is the stochastic gradient of $\bar{g}_i(\theta)$. Note that $\bar{g}_i(\theta)$ is the expectation of stochastic gradient $g_i(\theta_t)$ following the stationary distribution of environment $i$.

The detailed design of the HFTD algorithm is described as below (as summarized in Algorithm 1). In each round $t \in \{1, \dots, T\}$, the central server first broadcasts the global model $\bar{\theta}_t$ to all agents and each agent $i \in \{1, \dots, N\}$ independently performs $K_{i,t}$ local iterations starting from the current global model $\theta_i^{t,0}$ to optimize its local objective. $K_{i,t}$ may vary across agents since agents have *heterogeneous* computation capabilities. Following the same policy $\mu$, agent $i$ observes the tuple $O_i^{t,k} = (s_i^{t,k}, r_i^{t,k}, s_i^{t,k+1})$ at each local iteration $k$ of the round $t$ which is generated by its own MDP characterized by $\{\mathcal{S}, \mathcal{A}, \mathcal{P}_i, \mathcal{R}_i, \gamma\}$. Using the observation $O_i^{t,k}$, agent $i$ can compute the stochastic gradient by (2) and update its local model. At the end of each round, agents send the gradients directly to the server. The server then aggregates the gradients, updates the global model and starts round $t + 1$ of federation. Note that no exchange of raw samples is required, hence the privacy of local environments can be well protected. The update rule can be expressed as

$$\bar{\theta}^{t+1} = \bar{\theta}^t + \alpha \left( \frac{1}{N} \sum_{i=1}^N K_{i,t} \right) \cdot \frac{1}{N} \sum_{i=1}^N d_i^t \tag{7}$$

where $d_i^t$ is the normalized gradient for agent $i$ in the $t$-th round as $d_i^t = \frac{1}{K_{i,t}} \sum_{k=0}^{K_{i,t}-1} g_i(\theta_i^{t,k})$; $K_{i,t}$ is the number of local updates at agent $i$ in round $t$. Here we consider agents have heterogeneous number of local updates while the number of local updates are identical and fixed in Khodadadian et al. (2022); Wang et al. (2023); Jin et al. (2022). Note that cumulative local gradients are normalized when averaging, and this is a necessary technique when dealing with heterogeneous number of local updates in analysis [refer to Section 4.3 for details].

---

**Algorithm 1** Heterogeneous Federated TD (HFTD) Learning with Linear Function Approximation

1: **Input**: number of rounds $T$, step size $\alpha$, initial model $\theta_0$
2: **for** $t = 1$ to $T$ **do**
3:     $\theta_i^{t,0} \leftarrow \bar{\theta}^t$ for all agents $i$
4:     For each agent $i = 1, 2, ..., N$ do:
5:     **for** $k = 0$ to $K_{i,t} - 1$ **do**
6:         Observe a tuple $O_i^{t,k} = (s_i^{t,k}, r_i^{t,k}, s_i^{t,k+1})$ and calculate the gradient by (2)
7:         Update the local model by (3)
8:     **end for**
9:     Agents send the normalized gradient $d_i^t = \frac{\theta_i^{t,K_{i,t}-1} - \theta_i^{t,0}}{K_{i,t}}$ to the server
10:    Server computes the global model by (7)
11: **end for**
12: **Output**: $\{\bar{\theta}^t\}_{t=1}^T$

---

## 4.1 I.I.D. SAMPLING

First, we start from the scenario where the random tuples are independently and identically sampled from the stationary distribution $\pi_i$ of the Markov reward process for each agent $i$. That is to say, samples for updating the local model are independently drawn across iterations and rounds for each agent. We make the following assumptions, which are commonly imposed in federated reinforcement learning (Khodadadian et al., 2022; Wang et al., 2023; Fan et al., 2021).

**Assumption 1.** (**Bounded Gradient Variance**) For each agent $i$, there is a constant $\sigma$ such that $\mathbb{E}\|g_i(\theta) - \bar{g}_i(\theta)\| \leq \sigma^2$ for all $\theta \in \mathbb{R}^d$.

To measure the heterogeneity of environments, we have the following assumptions.

**Assumption 2.** (**Bounded Gradient Heterogeneity**) For any set of weights satisfying convex combination, i.e., $\{p_i \geq 0\}_{i=1}^m$ and $\sum_{i=1}^m p_i = 1$, there exist constants $\beta^2 \geq 1$, $\kappa^2 \geq 0$ such that $\sum_i p_i \|\bar{g}_i(\theta)\|_2^2 \leq \beta^2 \|\sum_i p_i \bar{g}_i(\theta)\|_2^2 + \kappa^2$. If agents execute in identical environments, then $\beta^2 = 1$, $\kappa^2 = 0$.

Assumption 2 is commonly used in the federated learning literature to capture the dissimilarities of local objectives.

**Assumption 3.** (**Smoothness**) Under the assumption that $\|\phi(s)\|_2^2 \leq 1$ for all $s$, it holds that $\|\bar{g}_i(\theta_1) - \bar{g}_i(\theta_2)\|_2 \leq 2\|\theta_1 - \theta_2\|_2$ for all $\theta_1, \theta_2 \in \mathbb{R}^d$.

Assumption 3 is important for deriving a fast convergence rate and the proof can be found in Bhandari et al. (2018); Tu & Recht (2019); Wang et al. (2023). Now we are ready to present the convergence guarantees using our HFTD algorithm:

**Theorem 1.** (**HFTD with I.I.D. Sampling**) Under Assumptions 1,2 and 3, let $K_{\max} = \max_{i,t}\{K_{i,t}\}$, $\bar{K}_{\max} = \max_t\{\bar{K}_t\}$ and $\bar{K}_{\min} = \min_t\{\bar{K}_t\}$. If $\alpha \leq \min_t\{\alpha_t\}$, Then output of Algorithm 1 can be represented as

$$\mathbb{E}\left\|\bar{\theta}^T - \theta^*\right\|_2^2 \leq e^{-\frac{\lambda\alpha\bar{K}_{\min}T}{8}}\left\|\bar{\theta}^0 - \theta^*\right\|_2^2 + \frac{256\alpha^2\sigma^2\bar{K}_{\max}(\bar{K}_{\max}-1)}{\lambda^2\bar{K}_{\min}}$$

$$+ \frac{512\alpha^2\kappa^2\bar{K}_{\max}K_{\max}(K_{\max}-1)}{\lambda^2\bar{K}_{\min}} + \frac{16\alpha\bar{K}_{\max}^2\sigma^2\hat{K}_{\min}}{N\lambda\bar{K}_{\max}}. \tag{8}$$

where $\hat{K}_t = \frac{1}{N} \sum_i \frac{1}{K_{i,t}}$ and $\hat{K}_{\max} = \min_t \{\hat{K}_t\}$.

**Remark 1.** Theorem 1 provides a bound on the convergence error of the HFTD algorithm when agents interact with heterogeneous environments using heterogeneous local iteration. The error bound consists of four terms. As $\alpha > 0$, the 1st term converges to zero as $T$ increases. Moreover, it achieves an exponential decay rate which matches the results of existing RL algorithms (Bhandari et al., 2018; Xu et al., 2020b; Kumar et al., 2023). The last three terms are all caused by the variances of stochastic gradients.

**Remark 2.** We note that the 4th term decreases as $N$ increases . Also note that the 3rd term becomes zero when each agent's local iteration number $K_{i,t}$ is 1 (i.e., perfect synchronization for all agents). Moreover, if agents interact with the same environments, then we have $\kappa^2 = 0$, and the 3rd term of the convergence bound vanishes.

**Remark 3.** We observe that the last three terms increase as the number of average local iteration goes up. This is because, due to agents' heterogeneous environments, more local iterations drive each agent's local model more towards its local optimal model and possibly away from the global optimal model (also known as "local drifts" in the existing works on federated supervised learning (Khaled et al., 2020; Wang et al., 2020).

We notice that the bound will be minimized when the numbers of local iterations are all equal to 1 (i.e., $K_{i,t} = 1, \forall i$). However, it has been widely shown that multiple local iterations achieve good performance empirically in practice while reducing communication costs (compared to communicating after every local iteration). Therefore, we should set local iteration numbers based on their empirical performance as well as agents' computation capabilities, rather than solely on the learning accuracy bound (which captures the worst-case scenario).

**Corollary 1.** Suppose a constant local update number $K$ for each agent, the convergence rate of HFTD with I.I.D. sampling is:

$$\mathbb{E} \left\| \bar{\theta}^T - \theta^* \right\|_2^2 \leq e^{-\frac{\lambda \alpha KT}{8}} \left\| \bar{\theta}^0 - \theta^* \right\|_2^2 + \frac{256\alpha^2\sigma^2(K-1)}{\lambda^2} + \frac{512\alpha^2\kappa^2 K(K-1)}{\lambda^2} + \frac{16\alpha\sigma^2}{\lambda N} \quad (9)$$

**Corollary 2.** Letting $\alpha = \sqrt{\frac{N}{KT}}$ yield

$$\mathbb{E} \left\| \bar{\theta}^T - \theta^* \right\|_2^2 \leq \mathcal{O} \left( e^{-\sqrt{NKT}} \right) + \mathcal{O} \left( \frac{N(K-1)}{T} \right) + \mathcal{O} \left( \frac{1}{\sqrt{NKT}} \right) \quad (10)$$

**Remark 4.** When the communication rounds $T$ is sufficiently large, then the convergence of HFTD will be dominated by the last term in (10) which is $\frac{1}{\sqrt{NKT}}$. Then we can conclude that the total complexity which can achieve an $\epsilon$-accurate optimal solution $\mathbb{E} \left\| \bar{\theta}^T - \theta^* \right\|_2^2 \leq \varepsilon$ is $KT = \mathcal{O} \left( \frac{1}{N\varepsilon^2} \right)$. When $K = 1$ and $N = 1$, the sample complexity will match the results in Theorem 2(b) in Bhandari et al. (2018)

## 4.2 MARKOVIAN SAMPLING

The case of I.I.D. sampling for RL can be hard to achieve in practical scenario. A more realistic setting is Markovian sampling, where the observed tuples used by TD are gathered from a single trajectory of the Markov chain. Different from the setting of I.I.D. sampling, Markovian sampling brings more challenges since samples are highly correlated. Specifically, in the I.I.D. case, $\mathbb{E}[g(\theta) - \bar{g}(\theta)] = 0$ since $g(\theta)$ is the unbiased estimate of $\bar{g}(\theta)$. However, in the Markovian setting, the samples for calculating $g(\theta)$ are not sampled from the stationary distribution. To put it another way, $\theta$ and the sample observed at time $t$, $O_t$, are not independent. Hence, $\mathbb{E}[g(\theta) - \bar{g}(\theta)] \neq 0$, indicating bias exists in the gradient evaluation for the analysis of a single agent. The analysis of federated temporal learning, will meet more complex time-correlations to deal with.

For the following analysis, we first introduce the geometric mixing property of finite-state, aperiodic and irreducible Markov chains as follows.

$$\sup \| P_i(x_k \in \cdot | x_0) - \pi_i(\cdot) \|_{TV} \leq \eta_i \rho_i^k \quad (11)$$

where $\pi(\cdot)$ is the stationary distribution of the MDP $i$; $\eta_i > 0$ and $\rho_i \in [0, 1]$ for all $i \in [N]$.

**Assumption 4.** (**Irreducibility and Aperiodicity**) For each $i \in [N]$, the Markov chain induced by the policy $\mu$, corresponding to the state transition matrix $\mathcal{P}_i$, is aperiodic and irreducible.

**Theorem 2.** (**HFTD with Markovian Sampling**) Under Assumptions 2, 3 and 4, if we choose $\alpha_t \leq \min\left\{ \frac{2}{\lambda(2c^2\bar{K}_t + c^2 + \bar{K}_t)}, \frac{1}{\lambda c'}, \frac{1}{4\left(\frac{\lambda}{24} + \frac{48}{\lambda} + 2\right)}, \frac{\lambda}{4\left(32\bar{K}_t^2\beta^2 c^2 + 16\bar{K}_t^2 + 16\beta^2 c^2 + \beta^2 c'\right)} \right\}$, then the output of Algorithm 1 satisfies

$$
\begin{aligned}
\mathbb{E}\left\|\bar{\theta}^T - \theta^*\right\|_2^2 \leq & e^{-\frac{\lambda \alpha \bar{K}_{\min} T}{4}} \mathbb{E}\left\|\bar{\theta}^0 - \theta^*\right\|_2^2 \\
& + \left(\frac{817}{\lambda^2} + \frac{8\tau^2 \alpha c}{\lambda}\right)\left(\frac{8\alpha^2 c^2 (K_{\max}-1)(2K_{\max}-1)}{3(1-C')}\left(\beta^2 H^2 + \kappa^2\right)\right) \\
& + \left(\frac{817}{\lambda^2} + \frac{8\tau^2 \alpha c}{\lambda}\right)\left(\frac{\alpha^2 q^2 (K_{\max}-1)(2K_{\max}-1)}{3(1-C')}\right) \\
& + \frac{8\alpha\left(2\bar{K}_{\max} + 2\tau^2 + 1\right)}{\lambda}\left(\frac{[q']^2 \hat{K}_{\max}}{N} + \frac{2qq'\rho(K_{\max}-1)\hat{K}_{\max}^2}{N(1-\rho)}\right) \\
& + \frac{8\alpha\left(2\bar{K}_{\max} + 2\tau^2 + 1\right)}{\lambda}\left(\frac{2[c']^2 \rho(K_{\max}-1)}{N^2(1-\rho)}\sum_{i<j}\frac{1}{K_{i,t}K_{j,t}}\rho^{\sum_{s=0}^{2\tau-1}(K_{i,s}+K_{j,s})}\right) \\
& \qquad\qquad\qquad\qquad\qquad\qquad\qquad\qquad\qquad\qquad\qquad\qquad\qquad\qquad\quad (12) \\
& + \frac{4\alpha\left(32c^2\bar{K}_{\max} + 16\bar{K}_{\max}c + 8\tau^2 c + c'\right)\kappa^2 + 2\alpha H\left(2c'H + q' + 4\beta^2\tau^2 cH\right)}{\lambda} \\
& \qquad\qquad\qquad\qquad\qquad\qquad\qquad\qquad\qquad\qquad\qquad\qquad\qquad\qquad\quad (13)
\end{aligned}
$$

where $\hat{K}_t = \frac{1}{N}\sum_i \frac{1}{K_{i,t}}$, $\hat{K}_{\max} = \max_t\{\hat{K}_t\}$, $\hat{K}_t^2 = \frac{1}{N}\sum_i \frac{1}{K_{i,t}^2}$, and $\hat{K}_{\max}^2 = \max_t\{\hat{K}_t^2\}$.

**Remark 5.** Theorem 2 characterizes the convergence of the HFTD algorithm where each agent's sampling follows a Markov chain. As in the setting of I.I.D. sampling, we can make similar observations here on the sampling complexity and the impacts of various system parameters on the convergence error. Different from the I.I.D. setting, we note that the term (12) comes from the variance reduction in the Markovian setting. If $t$ is sufficiently large, (12) diminishes to zero. This is because the Markov chain geometrically converges to its stationary distribution as $t$ evolves.

**Corollary 3.** If $N = 1$ and $K = 1$, then we have:

$$
\mathbb{E}\left\|\bar{\theta}^T - \theta^*\right\|_2^2 \leq e^{-\frac{\lambda \alpha T}{4}}\left\|\bar{\theta}^0 - \theta^*\right\|_2^2 + \frac{2\alpha H\left(2c'H + q' + 4\beta^2\tau^2 cH\right)}{\lambda} + \frac{8\alpha\left(3 + 2\tau^2\right)[q']^2}{\lambda} \tag{14}
$$

which matches the results of centralized TD as Theorem 3(b) in Bhandari et al. (2018).

### 4.3 TECHNICAL DIFFERENCES OF CONVERGENCE ANALYSIS

In this subsection, we highlight the key technical differences in the convergence analysis of the HFTD algorithm (i.e., the proofs of Theorems 1 and 2 and Corollary 1), compared to prior works.

In the convergence analysis, in order to bound $\mathbb{E}\left\|\bar{\theta}^{t+1} - \theta^*\right\|$, we need to bound an inner product term in (15), which can be decomposed into three terms. As the objective of the HFTD algorithm is to minimize the average MSPBE, the term $B$ can be *cancelled* (after the double summation before the inner product) due to the condition (6). In contrast, in Wang et al. (2023), this term $B$ cannot be cancelled and becomes a *non-vanishing bias* in the convergence error.

$$
\begin{aligned}
& \frac{1}{N}\sum_i \frac{1}{K_{i,t}}\sum_{k=0}^{K_{i,t}-1}\mathbb{E}\left\langle \bar{g}_i(\theta_i^{t,k}), \bar{\theta}^t - \theta^*\right\rangle \\
& = \frac{1}{N}\sum_i \frac{1}{K_{i,t}}\sum_{k=0}^{K_{i,t}-1}\mathbb{E}\left\langle \underbrace{\bar{g}_i(\theta_i^{t,k}) - \bar{g}_i(\bar{\theta}^t)}_{Assumption 3} + \underbrace{\bar{g}_i(\bar{\theta}^t) - \bar{g}(\bar{\theta}^t)}_{B} + \underbrace{\bar{g}(\bar{\theta}^t)}_{Lemma 3}, \bar{\theta}^t - \theta\right\rangle \tag{15}
\end{aligned}
$$

Similarly, in the convergence analysis of the Markovian setting, when dealing with such an inner product term, the term $B$ can also be cancelled (as shown in (27) in the Appendix).

In the convergence analysis, we also need to bound the error between the total accumulated local gradients $\frac{1}{N} \sum_i \frac{1}{K_{i,t}} \sum_{k=0}^{K_{i,t}-1} \bar{g}_i(\theta_i^{t,k})$ and the global gradient $\sum_i \bar{g}_i(\bar{\theta}^t)$ (as in (16)). By *normalizing* each agent's accumulated local gradients with the agent's local iteration number $K_{i,t}$, we are allowed to decompose the error into the sum of multiple error terms $\left\| \bar{g}_i(\theta_i^{t,k}) - \bar{g}_i(\bar{\theta}^t) \right\|$, each involving *only one* agent's local gradients. Then each of these error terms can be further bounded using the the smoothness condition of the local gradient.

$$
\mathbb{E} \left\| \frac{1}{N} \sum_i \frac{1}{K_{i,t}} \sum_{k=0}^{K_{i,t}-1} \bar{g}_i(\theta_i^{t,k}) - \bar{g}_i(\bar{\theta}^t) \right\|_2^2 \leq \frac{1}{N} \sum_i \mathbb{E} \left\| \frac{1}{K_{i,t}} \sum_{k=0}^{K_{i,t}-1} \bar{g}_i(\theta_i^{t,k}) - \bar{g}_i(\bar{\theta}^t) \right\|_2^2
$$

$$
\leq \frac{1}{N} \sum_i \frac{1}{K_{i,t}} \sum_{k=0}^{K_{i,t}-1} \mathbb{E} \left\| \bar{g}_i(\theta_i^{t,k}) - \bar{g}_i(\bar{\theta}^t) \right\|_2^2 \quad (16)
$$

## 5 SIMULATIONS

In this work, we have studied the problem of federated TD learning under heterogeneous environments with heterogeneous numbers of local updates. We provide numerical results under I.I.D. sampling setting and Markovian sampling setting on the platform: GridWorld. We first verify our theoretical results in a small-scale problem; see examples in Sutton & Barto (2018). Due to limited space, the details of the environment setting are provided in the Appendix.

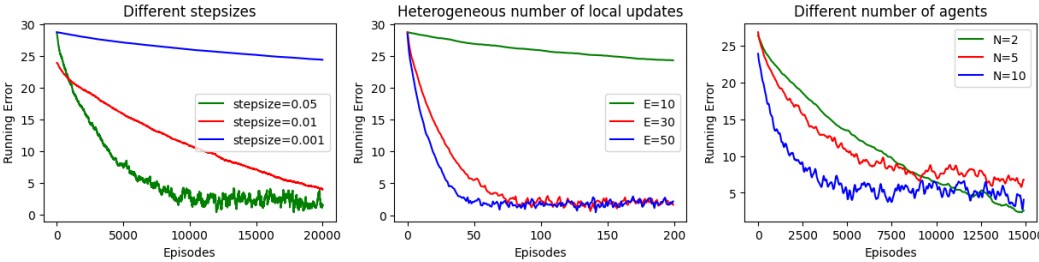

Figure 1: Training performance of HFTD with different settings under I.I.D. sampling: training performance refers to the running error between current model and optimal model. **Left:** agents perform local updates with different step sizes; **Middle:** agents take different number of local steps with mean values 10, 30, and 50; **Right:** the number of agents participating training is different.

Empirical results in Figure 1 reveal that the system parameters affect HFTD in different ways. Figure 1 (**Left**) confirms that HFTD with larger step size will lead to a faster convergence speed. However, it will have fluctuation when approaching to the optimal solution. As Theorem 1 shows, HFTD with smaller step size will lead the convergence error to zero. Figure 1 (**Middle**) shows that more local updates will lead to faster convergence. Figure 1 (**Right**) reveals that the federation of more agents will accelerate the learning process. However, due to the environment heterogeneity, the gap to optimal solution still exists unless we set a smaller step size.

## 6 CONCLUSION AND FUTURE WORK

In this paper, we have developed a HFTD algorithm for federated TD learning with linear function approximation under environment heterogeneity and computation heterogeneity. We have shown that the HFTD algorithm can asymptotically converge to the optimal value function model, which is the first such result in existing works on FRL with heterogeneous environments. The HFTD algorithm also achieves sampling complexity of $\mathcal{O}\left(\frac{1}{N\varepsilon^2}\right)$ and linear speedup that match the results of existing RL algorithms. For future work, we will study federated TD algorithms that allow for more flexibility for agents, including partial and asynchronous participation of agents. We will also explore FRL algorithms that involve both policy evaluation and policy improvement, such as the actor-critic algorithms.

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
