## 7 APPENDIX

In this appendix, we will give the proofs on I.I.D. case and Markovian case in Section H and J, respectively.

## H PROOF OF THEOREM 1 (I.I.D SETTING)

### H.1 PRELIMINARIES

By (2), the random TD(0) update direction of agent $i$ at the $t$-th communication round and $k$-th local iteration is:
$$g_i(\theta_i^{t,k}) = A_i(O_i^{t,k})\theta_i^{t,k} + b_i(O_i^{t,k})$$
where
$$A_i(O_i^{(t,k)}) = \phi(s_i^{(t,k)})(\gamma\phi(s_i^{(t,k+1)})^{\mathrm{T}} - \phi(s_i^{(t,k)})^{\mathrm{T}})$$
$$b_i(O_i^{(t,k)}) = r_i(s_i^{(t,k)})\phi(s_i^{(t,k)}).$$

For each agent $i$, let $\pi_i$ denote the stationary distribution of the associated Markov chain where agent $i$ is deployed at environment $i$ with transition probability $P^{(i)}$ under the fixed and common policy $\mu$; let $D_i$ be a diagonal matrix whose main diagonal consists of entries of $\pi^{(i)}$. $\theta_i^*$ is the unique solution of linear equation $\bar{A}_i\theta_i^* + \bar{b}_i = 0$ where $\bar{A}_i = \Phi^{\mathrm{T}}D_i(\gamma\mathcal{P}_i\Phi - \Phi)$ and $\bar{b}_i = \Phi^{\mathrm{T}}D_i\mathcal{R}_i$.

Note that the symbol with bar means the expectation of the physical value while the symbol with no bar means the physical value with randomness. The local expected gradient is:
$$\bar{g}_i(\theta) = \bar{A}_i\theta + \bar{b} = \bar{A}_i(\theta - \theta_i^*). \tag{17}$$

For the global objective $F(\theta) = \frac{1}{N}\sum\limits_{i=1}^{N}F_i(\theta)$, the global expected gradient is:
$$\bar{g}(\theta) = \frac{1}{N}\sum\limits_{i=1}^{N}\bar{g}_i(\theta) = \frac{1}{N}\sum\limits_{i=1}^{N}\bar{A}_i(\theta - \theta_i^*). \tag{18}$$

For the global objective, we define the following variables:

Normalized stochastic gradient:
$$d_i^{(t)} = \frac{1}{K_{i,t}}\sum\limits_{k=0}^{K_{i,t}-1}g_i(\theta_i^{t,k}) \tag{19}$$

Normalized gradient:
$$h_i^{(t)} = \frac{1}{K_{i,t}}\sum\limits_{k=0}^{K_{i,t}-1}\bar{g}_i(\theta_i^{t,k}) \tag{20}$$

Recall the update rule of the global model can be written as:
$$\bar{\theta}^{t+1} - \bar{\theta}^t = \left(\frac{1}{N}\sum\limits_{i=1}^{N}K_{i,t}\right)\cdot\frac{1}{N}\alpha\sum\limits_{i=1}^{N}d_i^{(t)} = \bar{K}_t\cdot\frac{1}{N}\sum\limits_{i=1}^{N}\alpha d_i^{(t)} \tag{21}$$
where $\bar{K}_t = \sum\limits_{i=1}^{N}p_iK_{i,t}$.

**Notations**

Let $\mathbb{E}_t$ denote the expectation considering on all the randomness up to the end of communication round $t-1$. Sometimes when there is no confusion, $\mathbb{E}$ represents $\mathbb{E}_t$. We also introduce $\mathcal{F}$ to represent the filtration. $\mathcal{F}_k^t$ represents the filtration that captures all the randomness up to $k$-th local iteration in round $t$; $\mathcal{F}^t$ represents the filtration up to round $t-1$.

**Lemma 1.** Suppose $\{M_t\}_{t=1}^T$ is a sequence random matrices and $E\left[M_t \,|\, M_{t-1}, M_{t-2}, \ldots M_1\right] = 0$, $\forall t$. Then,

$$\mathbb{E}\left[\left\|\sum_{t=1}^T M_t\right\|_F^2\right] = \sum_{t=1}^T \mathbb{E}\left[\|M_t\|_F^2\right].$$

*Proof.* The proof can be found in Wang et al. (2020). Assume $i < j$, it holds that $\mathbb{E}\left[M_i^{\mathrm{T}} M_j\right] = \mathbb{E}\left[M_i^{\mathrm{T}} \mathbb{E}\left[M_j \,|\, \mathcal{F}^i\right]\right] = 0$ □

**Lemma 2.** There exists a Markov chain whose expected gradient calculated by temporal difference learning is $\bar{g}(\theta)$. Then it holds that

$$\bar{g}(\theta) = \frac{1}{N}\sum_i \bar{g}_i(\theta) = \frac{1}{N}\sum_i \left(\Phi^{\mathrm{T}} D_i(\gamma P_i \Phi - \Phi)\theta + \Phi^{\mathrm{T}} D_i R_i\right)$$

$$= \Phi^{\mathrm{T}} D_* R_* + \Phi^{\mathrm{T}} D_*(\gamma P_* \Phi - \Phi)\theta$$

*Proof.* The sufficient condition is

$$\begin{cases} \frac{1}{N}\sum_i D_i R_i = D_* R_* \\ \frac{1}{N}\sum_i D_i(\gamma P_i - I) = D_*(\gamma P_* - I) \end{cases} \tag{22}$$

Where $I$ is the identity matrix. Then we can find one solution of (22) as

$$\begin{cases} R_* = \left(\sum_i D_i\right)^{-1}\left(\sum_i D_i R_i\right) \\ P_* = \left(\sum_i D_i\right)^{-1}\left(\sum_i D_i P_i\right) \end{cases}$$

where we assume $\sum_i D_i$ is invertible. Since $D_i$ is a diagonal matrix, it can be easily proved that $P_*$ is a transition matrix which is irreducible. □

Lemma 2 provides the basis for Lemma 3, which is important for deriving the convergence analysis.

**Lemma 3.** Assume that $\|\phi(s)\|_2^2 \leq 1$ for all $s$ and the columns of the feature matrix $\Phi$ are linearly independent. Then there exists a positive number $\lambda$ such that

$$\langle \bar{g}(\theta), \theta - \theta^* \rangle \leq -\frac{\lambda}{2}\|\theta - \theta^*\|_2^2.$$

*Proof.* The proof can be found in Bhandari et al. (2018); Xu et al. (2020a). □

To analyze the convergence performance of the proposed algorithm, we first characterize the estimation errors at each communication round.

**Per round progress:**

$$\mathbb{E}\left\|\bar{\theta}^{t+1} - \theta^*\right\|_2^2 = \mathbb{E}\left\|\bar{\theta}^t + \alpha_t \bar{K}_t \sum_i p_i d_i^{(t)} - \theta^*\right\|_2^2$$

$$= \mathbb{E}\left\|\bar{\theta}^t - \theta^*\right\|_2^2 + 2\alpha_t \bar{K}_t \underbrace{\mathbb{E}\left\langle \frac{1}{N}\sum_i h_i^{(t)}, \bar{\theta}^t - \theta^* \right\rangle}_{T_1} + \alpha_t^2 \bar{K}_t^2 \underbrace{\mathbb{E}\left\|\frac{1}{N}\sum_i d_i^{(t)}\right\|_2^2}_{T_2} \tag{23}$$

where the expectation is taken over $O_i^{t,k}$, $\forall i \in \{1, 2, \ldots m\}$, $\forall k \in \{0, 1, \ldots, K_{i,t} - 1\}$. The last equation is derived from Lemma 1 by letting $M_k = g_i(\theta_i^{t,k}) - \bar{g}_i(\theta_i^{t,k})$.

## H.2 BOUNDING THE SECOND TERM $T_1$

For the second term on the right hand side of (23), we have

$$
\begin{aligned}
T_1 &= \frac{1}{N} \sum_i \frac{1}{K_{i,t}} \sum_{k=0}^{K_{i,t}-1} \mathbb{E} \left\langle \bar{g}_i(\theta_i^{t,k}), \bar{\theta}^t - \theta^* \right\rangle \\
&= \frac{1}{N} \sum_i \frac{1}{K_{i,t}} \sum_{k=0}^{K_{i,t}-1} \mathbb{E} \left\langle \bar{g}_i(\theta_i^{t,k}) - \bar{g}_i(\bar{\theta}^t) + \bar{g}_i(\bar{\theta}^t) - \bar{g}(\bar{\theta}^t) + \bar{g}(\bar{\theta}^t), \bar{\theta}^t - \theta^* \right\rangle \\
&= \frac{1}{N} \sum_i \frac{1}{K_{i,t}} \sum_{k=0}^{K_{i,t}-1} \mathbb{E} \left\langle \bar{g}_i(\theta_i^{t,k}) - \bar{g}_i(\bar{\theta}^t), \bar{\theta}^t - \theta^* \right\rangle + \frac{1}{N} \sum_i \frac{1}{K_{i,t}} \sum_{k=0}^{K_{i,t}-1} \left\langle \bar{g}(\bar{\theta}^t), \bar{\theta}^t - \theta^* \right\rangle \\
&\leq \frac{1}{N} \sum_i \frac{1}{K_{i,t}} \sum_{k=0}^{K_{i,t}-1} \left( \frac{1}{2\xi_1} \mathbb{E} \left\| \bar{g}_i(\theta_i^{t,k}) - \bar{g}_i(\bar{\theta}^t) \right\|_2^2 + \frac{\xi_1}{2} \left\| \bar{\theta}^t - \theta^* \right\|_2^2 \right) + \left\langle \bar{g}(\bar{\theta}^t), \bar{\theta}^t - \theta^* \right\rangle
\end{aligned}
\tag{24}
$$

$$
\leq \frac{2}{\xi_1} \cdot \frac{1}{N} \sum_i \frac{1}{K_{i,t}} \sum_{k=0}^{K_{i,t}-1} \mathbb{E} \left\| \theta_i^{t,k} - \bar{\theta}^t \right\|_2^2 + \frac{\xi_1}{2} \left\| \bar{\theta}^t - \theta^* \right\|_2^2 + \left\langle \bar{g}(\bar{\theta}^t), \bar{\theta}^t - \theta^* \right\rangle
\tag{25}
$$

where (24) uses Young's inequality and $\xi_1$ can be any positive number; (25) follows from Assumption 3.

## H.3 BOUNDING THE THIRD TERM $T_2$

For the third term on the right hand side of (23), we have

$$
\begin{aligned}
T_2 &= \mathbb{E} \left\| \frac{1}{N} \sum_i d_i^{(t)} \right\|_2^2 \\
&= \mathbb{E} \left\| \frac{1}{N} \sum_i (d_i^{(t)} - h_i^{(t)}) + \frac{1}{N} \sum_i h_i^{(t)} \right\|_2^2 \\
&\leq 2\mathbb{E} \left\| \frac{1}{N} \sum_i (d_i^{(t)} - h_i^{(t)}) \right\|_2^2 + 2\mathbb{E} \left\| \frac{1}{N} \sum_i h_i^{(t)} \right\|_2^2 \\
&= 2\mathbb{E} \left\| \frac{1}{N} \sum_i \frac{1}{K_{i,t}} \sum_{k=0}^{K_{i,t}-1} \underbrace{g_i(\theta_i^{t,k}) - \bar{g}_i(\theta_i^{t,k})}_{M_i^{t,k}} \right\|_2^2 + 2\mathbb{E} \left\| \frac{1}{N} \sum_i h_i^{(t)} \right\|_2^2 \\
&= \frac{2}{N^2} \sum_i \frac{1}{K_{i,t}} \sum_{k=0}^{K_{i,t}-1} \mathbb{E} \left\| M_i^{t,k} \right\|_2^2 + 2\mathbb{E} \left\| \frac{1}{N} \sum_i h_i^{(t)} \right\|_2^2
\end{aligned}
\tag{26}
$$

$$
\leq \frac{2\sigma^2}{N^2} \sum_i \frac{1}{K_{i,t}} + 2\mathbb{E} \left\| \frac{1}{N} \sum_i h_i^{(t)} \right\|_2^2
\tag{27}
$$

where (26) is derived from Lemma 1 using the fact that $\mathbb{E} \left\langle M_i^{t,k}, M_{i'}^{t,k'} \right\rangle = 0$ when $i \neq i'$ or $k \neq k'$; (27) follows from Assumption 1.

Then, for the second term of (27), we have

$$\mathbb{E}\left\|\frac{1}{N}\sum_i h_i^{(t)}\right\|_2^2 = \mathbb{E}\left\|\frac{1}{N}\sum_i \frac{1}{K_{i,t}}\sum_{k=0}^{K_{i,t}-1}\bar{g}_i(\theta_i^{t,k})\right\|_2^2$$

$$= \mathbb{E}\left\|\frac{1}{N}\sum_i \frac{1}{K_{i,t}}\sum_{k=0}^{K_{i,t}-1}\bar{g}_i(\theta_i^{t,k}) - \bar{g}_i(\bar{\theta}^t) + \bar{g}_i(\bar{\theta}^t)\right\|_2^2$$

$$\leq 2\mathbb{E}\left\|\frac{1}{N}\sum_i \frac{1}{K_{i,t}}\sum_{k=0}^{K_{i,t}-1}\bar{g}_i(\theta_i^{t,k}) - \bar{g}_i(\bar{\theta}^t)\right\|_2^2 + 2\left\|\bar{g}(\bar{\theta}^t)\right\|_2^2$$

$$\leq \frac{2}{N}\sum_i \frac{1}{K_{i,t}}\sum_{k=0}^{K_{i,t}-1}\mathbb{E}\left\|\bar{g}_i(\theta_i^{t,k}) - \bar{g}_i(\bar{\theta}^t)\right\|_2^2 + 2\left\|\bar{g}(\bar{\theta}^t)\right\|_2^2 \tag{28}$$

$$\leq \frac{8}{N}\sum_i \frac{1}{K_{i,t}}\sum_{k=0}^{K_{i,t}-1}\mathbb{E}\left\|\theta_i^{t,k} - \bar{\theta}^t\right\|_2^2 + 2\left\|\bar{g}(\bar{\theta}^t)\right\|_2^2 \tag{29}$$

where (28) uses Jensen's Inequality: $\left\|\sum_{i=1}^N p_i z_i\right\|^2 \leq \sum_{i=1}^N p_i\|z_i\|^2$ while $\sum_i p_i = 1$ and $p_i$ is nonnegative for all agents; The last inequality follows from Assumption 3. Furthermore, $T_2$ satisfies

$$T_2 \leq 2\left(\frac{8}{N}\sum_i \frac{1}{K_{i,t}}\sum_{k=0}^{K_{i,t}-1}\mathbb{E}\left\|\theta_i^{t,k} - \bar{\theta}^t\right\|_2^2 + 2\left\|\bar{g}(\bar{\theta}^t)\right\|_2^2 + \frac{\sigma^2}{N^2}\sum_i \frac{1}{K_{i,t}}\right). \tag{30}$$

### H.4 Bounding the Drift Term

Considering the term $\frac{1}{N}\sum_i \frac{1}{K_{i,t}}\sum_{k=0}^{K_{i,t}-1}\mathbb{E}\left\|\theta_i^{t,k} - \bar{\theta}^t\right\|_2^2$, we have

$$\mathbb{E}\left\|\theta_i^{t,k} - \bar{\theta}^t\right\|_2^2 = \alpha_t^2 \mathbb{E}\left\|\sum_{s=0}^{k-1} g_i(\theta_i^{t,s})\right\|_2^2$$

$$\leq 2\alpha_t^2 \mathbb{E}\left\|\sum_{s=0}^{k-1}(g_i(\theta_i^{t,s}) - \bar{g}_i(\theta_i^{t,s}))\right\|_2^2 + 2\alpha_t^2 \mathbb{E}\left\|\sum_{s=0}^{k-1}\bar{g}_i(\theta_i^{t,s})\right\|_2^2$$

$$= 2\alpha_t^2 \sum_{s=0}^{k-1}\mathbb{E}\left\|g_i(\theta_i^{t,s}) - \bar{g}_i(\theta_i^{t,s})\right\|_2^2 + 2\alpha_t^2 \mathbb{E}\left\|\sum_{s=0}^{k-1}\bar{g}_i(\theta_i^{t,s})\right\|_2^2$$

$$\leq 2\alpha_t^2 \sigma^2 k + 2\alpha_t^2 \mathbb{E}\left\|\sum_{s=0}^{k-1}\bar{g}_i(\theta_i^{t,s})\right\|_2^2$$

$$\leq 2\alpha_t^2 \sigma^2 k + 2\alpha_t^2 k\sum_{s=0}^{k-1}\mathbb{E}\left\|\bar{g}_i(\theta_i^{t,s})\right\|_2^2. \tag{31}$$

where (31) is derived from Jensen's inequality. Note that

$$\frac{1}{K_{i,t}}\sum_{k=0}^{K_{i,t}-1}k = \frac{K_{i,t}-1}{2}. \tag{32}$$

Then we have:

$$\frac{1}{K_{i,t}}\sum_{k=0}^{K_{i,t}-1}\mathbb{E}\left\|\theta_i^{t,k} - \bar{\theta}^t\right\|_2^2 \leq \alpha_t^2 \sigma^2(K_{i,t}-1) + \alpha_t^2(K_{i,t}-1)\sum_{k=0}^{K_{i,t}-1}\mathbb{E}\left\|\bar{g}_i(\theta_i^{t,k})\right\|_2^2. \tag{33}$$

In addition, we can bound the second term as:

$$\mathbb{E}\left\|\bar{g}_i(\theta_i^{t,k})\right\|_2^2 \le 2\mathbb{E}\left\|\bar{g}_i(\theta_i^{t,k}) - \bar{g}_i(\bar{\theta}^t)\right\|_2^2 + 2\mathbb{E}\left\|\bar{g}_i(\bar{\theta}^t)\right\|_2^2$$

$$\le 8\mathbb{E}\left\|\theta_i^{t,k} - \bar{\theta}^t\right\|_2^2 + 2\mathbb{E}\left\|\bar{g}_i(\bar{\theta}^t)\right\|_2^2 \tag{34}$$

Substituting (34) into (33), we have

$$\frac{1}{K_{i,t}} \sum_{k=0}^{K_{i,t}-1} \mathbb{E}\left\|\theta_i^{t,k} - \bar{\theta}^t\right\|_2^2 \le \alpha_t^2\sigma^2(K_{i,t} - 1) + 8\alpha_t^2(K_{i,t} - 1)\sum_{k=0}^{K_{i,t}-1}\mathbb{E}\left\|\theta_i^{t,k} - \bar{\theta}^t\right\|_2^2$$

$$+ 2\alpha_t^2(K_{i,t} - 1)^2\mathbb{E}\left\|\bar{g}_i(\bar{\theta}^t)\right\|_2^2$$

After minor rearranging, it follows that

$$\frac{1}{K_{i,t}} \sum_{k=0}^{K_{i,t}-1} \mathbb{E}\left\|\theta_i^{t,k} - \bar{\theta}^t\right\|_2^2 \le \frac{\alpha_t^2\sigma^2(K_{i,t} - 1)}{1 - 8\alpha_t^2 K_{i,t}(K_{i,t} - 1)} + \frac{2\alpha_t^2(K_{i,t} - 1)^2\mathbb{E}\left\|\bar{g}_i(\bar{\theta}^t)\right\|_2^2}{1 - 8\alpha_t^2 K_{i,t}(K_{i,t} - 1)}. \tag{35}$$

Define $C_t = \max_i\left\{8\alpha_t^2 K_{i,t}(K_{i,t} - 1)\right\} \le 1$, then (35) can be simplified as

$$\frac{1}{K_{i,t}} \sum_{k=0}^{K_{i,t}-1} \mathbb{E}\left\|\theta_i^{t,k} - \bar{\theta}^t\right\|_2^2 \le \frac{\alpha_t^2\sigma^2(K_{i,t} - 1)}{1 - C_t} + \frac{C_t}{4(1 - C_t)}\mathbb{E}\left\|\bar{g}_i(\bar{\theta}^t)\right\|_2^2$$

In summary, the client drift can be expressed as

$$\frac{1}{N}\sum_i \frac{1}{K_{i,t}} \sum_{k=0}^{K_{i,t}-1} \mathbb{E}\left\|\theta_i^{t,k} - \bar{\theta}^t\right\|_2^2 \le \frac{\alpha_t^2\sigma^2(\bar{K}_t - 1)}{1 - C_t} + \frac{C_t}{4(1 - C_t)N}\sum_i\left\|\bar{g}_i(\bar{\theta}^t)\right\|_2^2$$

$$\le \frac{\alpha_t^2\sigma^2(\bar{K}_t - 1)}{1 - C_t} + \frac{C_t\beta^2}{4(1 - C_t)}\left\|\bar{g}(\bar{\theta}^t)\right\|_2^2$$

$$+ \frac{C_t\kappa^2}{4(1 - C_t)} \tag{36}$$

where (36) follows from Assumption 2, and $\bar{K}_t = \frac{1}{N}\sum_i K_{i,t}$.

## H.5 FINAL RESULTS

Inspired by Bhandari et al. (2018), we can notice that there exists a positive constant number $\lambda$ such that

$$\langle\bar{g}(\theta), \theta - \theta^*\rangle \le -\frac{\lambda}{2}\|\theta - \theta^*\|_2^2.$$

Besides, we can get

$$\|\bar{g}(\theta)\|_2 \le 2\|\theta - \theta^*\|_2.$$

By combining the intermediate results on $T_1$ and $T_2$, the results of *per round progress* can be rewritten as

$$\mathbb{E}\left\|\bar{\theta}^{t+1} - \theta^*\right\|_2^2 = \mathbb{E}\left\|\bar{\theta}^t - \theta^*\right\|_2^2 + 2\alpha_t\bar{K}_t\mathbb{E}\left\langle\frac{1}{N}\sum_i h_i^{(t)}, \bar{\theta}^t - \theta^*\right\rangle + \alpha_t^2\bar{K}_t^2\mathbb{E}\left\|\frac{1}{N}\sum_i d_i^{(t)}\right\|_2^2$$

$$\le (1 + \alpha_t\bar{K}_t\xi_1)\mathbb{E}\left\|\bar{\theta}^t - \theta^*\right\|_2^2 + \frac{4\alpha_t\bar{K}_t}{\xi_1}\cdot\frac{1}{N}\sum_i\frac{1}{K_{i,t}}\sum_{k=0}^{K_{i,t}-1}\left\|\theta_i^{t,k} - \bar{\theta}^t\right\|_2^2 + 2\alpha_t\bar{K}_t\mathbb{E}\left\langle\bar{g}(\bar{\theta}^t), \bar{\theta}^t - \theta^*\right\rangle$$

$$+ 2\alpha_t^2\bar{K}_t^2\left(\frac{8}{N}\sum_i\frac{1}{K_{i,t}}\sum_{k=0}^{K_{i,t}-1}\mathbb{E}\left\|\theta_i^{t,k} - \bar{\theta}^t\right\|_2^2 + 2\left\|\bar{g}(\bar{\theta}^t)\right\|_2^2 + \frac{\sigma^2}{N^2}\sum_i\frac{1}{K_{i,t}}\right).$$

Plugging (36) into the last inequality, we can get:

$$
\mathbb{E} \left\| \bar{\theta}^{t+1} - \theta^* \right\|_2^2 \leq (1 + \alpha_t \bar{K}_t \xi_1) \mathbb{E} \left\| \bar{\theta}^t - \theta^* \right\|_2^2 + 4\alpha_t^2 \bar{K}_t^2 \left\| \bar{g}(\bar{\theta}^t) \right\|_2^2
$$
$$
+ \left( \frac{4\alpha_t \bar{K}_t}{\xi_1} + 16\alpha_t^2 \bar{K}_t^2 \right) \left( \frac{\alpha_t^2 \sigma^2 (\bar{K}_t - 1)}{1 - C_t} + \frac{C_t \beta^2}{4(1 - C_t)} \left\| \bar{g}(\bar{\theta}^t) \right\|_2^2 + \frac{C_t \kappa^2}{4(1 - C_t)} \right)
$$
$$
+ 2\alpha_t \bar{K}_t \mathbb{E} \left\langle \bar{g}(\bar{\theta}^t), \bar{\theta}^t - \theta^* \right\rangle + \frac{2\alpha_t^2 \bar{K}_t^2 \sigma^2}{N^2} \sum_i \frac{1}{K_{i,t}}.
$$

Applying Lemma 3, we have:

$$
\mathbb{E} \left\| \bar{\theta}^{t+1} - \theta^* \right\|_2^2 \leq \left( 1 + \alpha_t \bar{K}_t \xi_1 - \lambda \alpha_t \bar{K}_t + 16\alpha_t^2 \bar{K}_t^2 + \left( \frac{4\alpha_t \bar{K}_t}{\xi_1} + 16\alpha_t^2 \bar{K}_t^2 \right) \frac{C_t \beta^2}{1 - C_t} \right) \left\| \bar{\theta}^t - \theta^* \right\|_2^2
$$
$$
+ \left( \frac{4\alpha_t \bar{K}_t}{\xi_1} + 16\alpha_t^2 \bar{K}_t^2 \right) \frac{\alpha_t^2 \sigma^2 (\bar{K}_t - 1)}{1 - C_t} + \left( \frac{\alpha_t \bar{K}_t}{\xi_1} + 4\alpha_t^2 \bar{K}_t^2 \right) \frac{C_t \kappa^2}{1 - C_t} + \frac{2\alpha_t^2 \bar{K}_t^2 \sigma^2}{N^2} \sum_i \frac{1}{K_{i,t}}.
$$
$$
\tag{37}
$$

Let $\xi_1 = \frac{\lambda}{2}$, then (37) can be represented as

$$
\mathbb{E} \left\| \bar{\theta}^{t+1} - \theta^* \right\|_2^2 \leq \left( 1 - \frac{\lambda \alpha_t \bar{K}_t}{2} + 16\alpha_t^2 \bar{K}_t^2 + \left( \frac{8\alpha_t \bar{K}_t}{\lambda} + 16\alpha_t^2 \bar{K}_t^2 \right) \frac{C_t \beta^2}{1 - C_t} \right) \left\| \bar{\theta}^t - \theta^* \right\|_2^2
$$
$$
+ \left( \frac{8\alpha_t \bar{K}_t}{\lambda} + 16\alpha_t^2 \bar{K}_t^2 \right) \frac{\alpha_t^2 \sigma^2 (\bar{K}_t - 1)}{1 - C_t} + \left( \frac{2\alpha_t \bar{K}_t}{\lambda} + 4\alpha_t^2 \bar{K}_t^2 \right) \frac{C_t \kappa^2}{1 - C_t} + \frac{2\alpha_t^2 \bar{K}_t^2 \sigma^2}{N^2} \sum_i \frac{1}{K_{i,t}}.
$$
$$
\tag{38}
$$

If $\alpha_t \leq \frac{\lambda}{128 \bar{K}_t}$, then inequalities $\frac{8\alpha_t \bar{K}_t}{\lambda} + 16\alpha_t^2 \bar{K}_t^2 \leq \frac{16\alpha_t \bar{K}_t}{\lambda}$ and $16\alpha_t^2 \bar{K}_t^2 \leq \frac{\lambda \alpha_t \bar{K}_t}{8}$ hold. If $C_t \leq \frac{\lambda^2}{64\beta^2 + \lambda^2}$, it follows that $\frac{1}{1 - C_t} \leq 1 + \frac{\lambda^2}{64\beta^2}$ and $\frac{C_t \beta^2}{1 - C_t} \leq \frac{\lambda^2}{64}$. Recall that $C_t = \max_i \left\{ 8\alpha_t^2 K_{i,t}(K_{i,t} - 1) \right\} \leq 1$. We denote $K_{t,\max} = \max_i \{K_{i,t}\}$ then $C_t$ can be rewritten as $C_t = 8\alpha_t^2 K_{t,\max}(K_{t,\max} - 1)$. These facts can help us further simplify inequality (38) as:

$$
\mathbb{E} \left\| \bar{\theta}^{t+1} - \theta^* \right\|_2^2 \leq \left( 1 - \frac{\lambda \alpha_t \bar{K}_t}{8} \right) \left\| \bar{\theta}^t - \theta^* \right\|_2^2 + \frac{32\alpha_t^3 \sigma^2 \bar{K}_t (\bar{K}_t - 1)}{\lambda}
$$
$$
+ \frac{64\alpha_t^3 \kappa^2 \bar{K}_t K_{t,\max}(K_{t,\max} - 1)}{\lambda} + \frac{2\alpha_t^2 \bar{K}_t^2 \sigma^2}{N^2} \sum_i \frac{1}{K_{i,t}}.
\tag{39}
$$

Then let $K_{\max} = \max_{i,t} \{K_{i,t}\}$, $\bar{K}_{\max} = \max_t \{\bar{K}_t\}$ and $\bar{K}_{\min} = \min_t \{\bar{K}_t\}$. If $\alpha \leq \min_t \{\alpha_t\}$, applying (39) recursively from $t = 0$ to $T - 1$, we have:

$$
\mathbb{E} \left\| \bar{\theta}^T - \theta^* \right\|_2^2 \leq e^{-\frac{\lambda \alpha \bar{K}_{\min} T}{8}} \left\| \bar{\theta}^0 - \theta^* \right\|_2^2 + \frac{256\alpha^2 \sigma^2 \bar{K}_{\max}(\bar{K}_{\max} - 1)}{\lambda^2 \bar{K}_{\min}}
$$
$$
+ \frac{512\alpha^2 \kappa^2 \bar{K}_{\max} K_{\max}(K_{\max} - 1)}{\lambda^2 \bar{K}_{\min}} + \frac{16\alpha \bar{K}_{\max}^2 \sigma^2 \hat{K}_{\max}}{N \lambda \bar{K}_{\min}}.
\tag{40}
$$

where $\hat{K}_t = \frac{1}{N} \sum_i \frac{1}{K_{i,t}}$ and $\hat{K}_{\max} = \max_t \{\hat{K}_t\}$.

## I  PROOF OF COROLLARY 1

Assuming we use a constant local update number $K$ for each agent, (39) can be simplified as

$$
\mathbb{E} \left\| \bar{\theta}^{t+1} - \theta^* \right\|_2^2 \leq \left( 1 - \frac{\lambda \alpha K}{8} \right) \left\| \bar{\theta}^t - \theta^* \right\|_2^2 + \frac{32\alpha^2 \sigma^2 K(K - 1)}{\lambda} + \frac{64\alpha^2 \kappa^2 K^2(K - 1)}{\lambda} + \frac{2\alpha^2 K \sigma^2}{N}.
\tag{41}
$$

Then we have:

$$\mathbb{E}\left\|\bar{\theta}^T - \theta^*\right\|_2^2 \leq e^{-\frac{\lambda\alpha KT}{8}}\left\|\bar{\theta}^0 - \theta^*\right\|_2^2 + \frac{256\alpha^2\sigma^2(K-1)}{\lambda^2} + \frac{512\alpha^2\kappa^2 K(K-1)}{\lambda^2} + \frac{16\alpha\sigma^2}{\lambda N}. \tag{42}$$

Letting $\alpha = \sqrt{\frac{N}{KT}}$ yields

$$\mathbb{E}\left\|\bar{\theta}^T - \theta^*\right\|_2^2 \leq \mathcal{O}\left(e^{-\sqrt{NKT}}\right) + \mathcal{O}\left(\frac{N(K-1)}{T}\right) + \mathcal{O}\left(\frac{1}{\sqrt{NKT}}\right). \tag{43}$$

## J    PROOF OF THEOREM 2 (MARKOVIAN SETTING)

Inspired by Sun et al. (2020); Khodadadian et al. (2022); Wang et al. (2023), we first decompose the stochastic gradient as

$$g_i(\theta_i^{t,k}) = A_i(O_i^{t,k})(\theta_i^{t,k} - \theta_i^*) + \underbrace{A_i(O_i^{t,k})\theta_i^* + b_i\left(O_i^{t,k}\right)}_{Z_i(O_i^{t,k})}$$

From the analysis in I.I.D. setting, we know that the expected gradient can be written as

$$\bar{g}_i(\theta_i^{t,k}) = A_i(O_i^{t,k})(\theta_i^{t,k} - \theta_i^*) \tag{44}$$

where $\bar{A}_i = \Phi^{\mathrm{T}} D^{(i)}(\gamma P^{(i)}\Phi - \Phi)$. At the same time, $\mathbb{E}_{O_i^{t,k}\sim\pi_i}\left[Z_i(O_i^{t,k})\right] = 0$ holds.

In the following, We first give the uniform bound of these variables.

**Lemma 4.** (**Bounded Variables**)
Given $\|\phi(s)\|_2^2 \leq 1$, we can bound the following variables as $\left\|A_i(O_i^{t,k})\right\|_2 \leq 1+\gamma$, $\left\|\bar{A}_i(O_i^{t,k})\right\|_2 \leq 1+\gamma$ and $\left\|Z_i(O_i^{t,k})\right\|_2 \leq (1+\gamma)H + R$ hold for all $i \in [N]$, where $H$ denotes the radius of the set which includes all local optimal solutions and global solution and $R$ denotes the upper bound of the reward.

*Proof.* Since $\|\phi(s)\|_2^2 \leq 1$, we can conclude that

$$\left\|A_i(O_i^{t,k})\right\|_2 = \left\|\phi(s_i^{t,k})(\gamma\phi^{\mathrm{T}}(s_i^{t,k}) - \phi^{\mathrm{T}}(s_i^{t,k+1}))\right\|_2 \leq \left\|\phi(s_i^{t,k})\right\|_2 \left\|\gamma\phi^{\mathrm{T}}(s_i^{t,k}) - \phi^{\mathrm{T}}(s_i^{t,k+1})\right\|_2 \leq c$$

where $c := 1 + \gamma$. Then using the triangle inequality, we have:

$$\left\|Z_i(O_i^{t,k})\right\|_2 = \left\|A_i(O_i^{t,k})\theta_i^* + b_i(O_i^{t,k})\right\|_2 \leq \left\|A_i(O_i^{t,k})\theta_i^*\right\|_2 + \left\|b_i(O_i^{t,k})\right\|_2 \leq cH + R \tag{45}$$

$\square$

Since there are strong dependencies between the noisy observations in the Markovian chain model (Bhandari et al., 2018) , we need to compute the bias between the expectation of stochastic $A_i$ and $\bar{A}_i$ which is zero in I.I.D. sampling, and the bias of the expectation of $Z_i$ which is also zero in I.I.D. sampling.

**Lemma 5.** (**Bias of Variables' Expectation**)

$$\left\|\bar{A}_i - \mathbb{E}\left[A_i(O_i^{t_2,k_2})\,\middle|\,F_{k_1}^{t_1}\right]\right\|_2 \leq 2c\eta_i\rho_i^{\sum_{t=t_1}^{t_2-1} K_{i,t}+k_2-k_1} \leq c'\rho_i^{\sum_{t=t_1}^{t_2-1} K_{i,t}+k_2-k_1}$$

where $c' = 2c\max_i\{\eta_i\}$

Similarly, for the bias of the expectation of $Z_i$ using Markovian sampling, we have

$$\left\|\mathbb{E}\left[Z_i(O_i^{t_2,k_2})\,\middle|\,F_{k_1}^{t_1}\right]\right\|_2 \leq 2q\eta_i\rho_i^{\sum_{t=t_1}^{t_2-1} K_{i,t}+k_2-k_1} \leq q'\rho_i^{\sum_{t=t_1}^{t_2-1} K_{i,t}+k_2-k_1}$$

where $q := cH + R$; $q' = 2q\max_i\{\eta_i\}$. The proof can be found in Sun et al. (2020); Wang et al. (2023).

### J.1    VARIANCE REDUCTION

In this section, we give the bound of the gradient variance. Since the gradient can be decomposed into two terms, we discuss them separately.

$$\mathbb{E}_{t-\tau}\left\|\frac{1}{N}\sum_i\frac{1}{K_{i,t}}\sum_{k=0}^{K_{i,t}-1}Z_i(O_i^{t,k})\right\|_2^2$$

$$=\mathbb{E}_{t-\tau}\left(\frac{1}{N}\sum_i\frac{1}{K_{i,t}}\sum_{k=0}^{K_{i,t}-1}Z_i(O_i^{t,k})\right)^{\mathrm{T}}\left(\frac{1}{N}\sum_i\frac{1}{K_{i,t}}\sum_{k=0}^{K_{i,t}-1}Z_i(O_i^{t,k})\right)$$

$$=\frac{1}{N^2}\mathbb{E}_{t-\tau}\left[\sum_i\frac{1}{K_{i,t}^2}\sum_{k=0}^{K_{i,t}-1}Z_i^{\mathrm{T}}(O_i^{t,k})Z_i^{\mathrm{T}}(O_i^{t,k})+2\sum_i\frac{1}{K_{i,t}^2}\sum_{k<l}Z_i^{\mathrm{T}}(O_i^{t,k})Z_i(O_i^{t,l})\right.$$

$$\left.+2\sum_{i<j}\frac{p_ip_j}{K_{i,t}K_{j,t}}\sum_{k\in[K_{i,t}],l\in[K_{j,t}]}Z_i^{\mathrm{T}}(O_i^{t,k})Z_j(O_j^{t,l})\right]\qquad(46)$$

For the second term of (46),

$$\mathbb{E}_{t-\tau}\left[2\sum_i\frac{1}{K_{i,t}^2}\sum_{k<l}Z_i^{\mathrm{T}}(O_i^{t,k})Z_i(O_i^{t,l})\right]$$

$$=\mathbb{E}_{t-\tau}\left[2\sum_i\frac{1}{K_{i,t}^2}\sum_{k<l}Z_i^{\mathrm{T}}(O_i^{t,k})\mathbb{E}\left[Z_i(O_i^{t,l})\,|F_t^k\right]\right]$$

$$\leq\mathbb{E}_{t-\tau}\left[2qq'\sum_i\frac{1}{K_{i,t}^2}\sum_{k<l}\rho^{l-k}\right]$$

where the last inequality is derived from Lemma 4 and Lemma 5. Then it follows:

$$\mathbb{E}_{t-\tau}\left[2\sum_i\frac{1}{K_{i,t}^2}\sum_{k<l}Z_i^{\mathrm{T}}(O_i^{t,k})Z_i(O_i^{t,l})\right]\leq\mathbb{E}_{t-\tau}\left[2qq'\sum_i\frac{1}{K_{i,t}^2}\sum_{k=0}^{K_{i,t}-1}\sum_{l=k+1}^{K_{i,t}-1}\rho^{l-k}\right]$$

$$\leq\mathbb{E}_{t-\tau}\left[\frac{2qq'\rho}{1-\rho}\sum_i\frac{K_{i,t}-1}{K_{i,t}^2}\right]\qquad(47)$$

$$\leq\mathbb{E}_{t-\tau}\left[\frac{2qq'\rho(K_{t,\max}-1)}{1-\rho}\sum_i\frac{1}{K_{i,t}^2}\right]$$

where $\rho=\max_i\{\rho_i\}$, $K_{t,\max}=\max_i\{K_{i,t}\}$ and (47) is derived from the fact that $\sum_{a=1}^{\infty}\rho^a=\frac{\rho}{1-\rho}$ when $0<\rho<1$. Now we study the third term of (46) where observation of different agents are mutually independent.

For the third term,

$$\mathbb{E}_{t-\tau}\left[2\sum_{i<j}\frac{1}{K_{i,t}K_{j,t}}\sum_{k\in[K_{i,t}],l\in[K_{j,t}]}Z_i^{\mathrm{T}}(O_i^{t,k})Z_j(O_j^{t,l})\right]$$

$$\leq2\sum_{i<j}\frac{1}{K_{i,t}K_{j,t}}\sum_{k\in[K_{i,t}],l\in[K_{j,t}]}\mathbb{E}_{t-\tau}\left\|Z_i^{\mathrm{T}}(O_i^{(t,j)})\right\|_2\mathbb{E}_{t-\tau}\left\|Z_k(O_k^{(t,l)}))\right\|_2$$

$$=2[q']^2\sum_{i<j}\frac{1}{K_{i,t}K_{j,t}}\sum_{k\in[K_{i,t}],l\in[K_{j,t}]}\rho^{\sum_{p=t-\tau}^{t-1}(K_{i,p}+K_{j,p})+k+l}$$

where the last inequality is derived from Lemma '5. Then the third term can be further bounded as

$$
\mathbb{E}_{t-\tau}\left[2\sum_{i<j}\frac{1}{K_{i,t}K_{j,t}}\sum_{k\in[K_{i,t}],l\in[K_{j,t}]}Z_i^{\mathrm{T}}(O_i^{t,k})Z_j(O_j^{t,l})\right]
$$

$$
=2[q']^2\sum_{i<j}\frac{1}{K_{i,t}K_{j,t}}\rho^{\tau(K_{i,t}+K_{j,t})}\sum_{k\in[K_{i,t}],l\in[K_{j,t}]}\rho^{k+l}
$$

$$
\leq\frac{2[q']^2\rho(K_{t,\max}-1)}{1-\rho}\sum_{i<j}\frac{1}{K_{i,t}K_{j,t}}\rho^{\sum_{p=t-\tau}^{t-1}(K_{i,p}+K_{j,p})}
$$

Then (46) can be represented as

$$
\mathbb{E}_{t-\tau}\left\|\frac{1}{N}\sum_i\frac{1}{K_{i,t}}\sum_{k=0}^{K_{i,t}-1}Z_i(O_i^{t,k})\right\|_2^2\leq\frac{1}{N^2}\sum_i\frac{[q']^2}{K_{i,t}}+\frac{2qq'\rho(K_{t,\max}-1)}{N^2(1-\rho)}\sum_i\frac{1}{K_{i,t}^2}
$$

$$
+\frac{2[q']^2\rho(K_{t,\max}-1)}{N^2(1-\rho)}\sum_{i<j}\frac{1}{K_{i,t}K_{j,t}}\rho^{\sum_{p=t-\tau}^{t-1}(K_{i,p}+K_{j,p})}
$$

$$
\tag{48}
$$

Similarly, we have

$$
\mathbb{E}_{t-\tau}\left\|\frac{1}{N}\sum_i\frac{1}{K_{i,t}}\sum_{k=0}^{K_{i,t}-1}A_i(O_i^{t,k})-\bar{A}_i\right\|_2^2\leq\frac{1}{N^2}\sum_i\frac{1[c']^2}{K_{i,t}}+\frac{2cc'\rho(K_{t,\max}-1)}{N^2(1-\rho)}\sum_i\frac{1}{K_{i,t}^2}
$$

$$
+\frac{2[c']^2\rho(K_{t,\max}-1)}{N^2(1-\rho)}\sum_{i<j}\frac{1}{K_{i,t}K_{j,t}}\rho^{\sum_{p=t-\tau}^{t-1}(K_{i,p}+K_{j,p})}
$$

$$
\tag{49}
$$

## J.2 PER ROUND PROGRESS IN MARKOVIAN SAMPLING

In this section, we will analyze the convergence performance under Markovian sampling. For any $t>\tau$, we have

$$
\mathbb{E}_{t-\tau}\left\|\bar{\theta}^{t+1}-\theta^*\right\|_2^2
$$

$$
=\mathbb{E}_{t-\tau}\left\|\bar{\theta}^t+\alpha\bar{K}_t\cdot\frac{1}{N}\sum_i d_i^{(t)}-\theta^*\right\|_2^2
$$

$$
=\mathbb{E}_{t-\tau}\left\|\bar{\theta}^t-\theta^*\right\|_2^2+2\alpha\bar{K}_t\underbrace{\mathbb{E}_{t-\tau}\left\langle\frac{1}{N}\sum_i h_i^{(t)},\bar{\theta}^t-\theta^*\right\rangle}_{W_1}+\alpha^2\bar{K}_t^2\underbrace{\mathbb{E}_{t-\tau}\left\|\frac{1}{N}\sum_i d_i^{(t)}\right\|_2^2}_{W_2}
$$

$$
+2\alpha\bar{K}_t\underbrace{\mathbb{E}_{t-\tau}\left\langle\frac{1}{N}\sum_i(d_i^{(t)}-h_i^{(t)}),\bar{\theta}^t-\theta^*\right\rangle}_{W_3}
$$

$$
\tag{50}
$$

where $W_3$ is a biased term due to Markovian sampling.

For the second term $W_1$ on the right hand side in (50), we can obtain:

$$
W_1 = \mathbb{E}_{t-\tau} \left\langle \frac{1}{N} \sum_i h_i^{(t)}, \bar{\theta}^t - \theta^* \right\rangle = \mathbb{E}_{t-\tau} \left\langle \frac{1}{N} \sum_i \frac{1}{K_{i,t}} \sum_{k=0}^{K_{i,t}-1} \bar{g}_i(\theta_i^{t,k}), \bar{\theta}^t - \theta^* \right\rangle
$$

$$
= \mathbb{E}_{t-\tau} \left\langle \frac{1}{N} \sum_i \frac{1}{K_{i,t}} \sum_{k=0}^{K_{i,t}-1} (\bar{g}_i(\theta_i^{t,k}) - \bar{g}_i(\bar{\theta}^t) + \bar{g}_i(\bar{\theta}^t) - \bar{g}(\bar{\theta}^t) + \bar{g}(\bar{\theta}^t)), \bar{\theta}^t - \theta^* \right\rangle
$$

$$
= \mathbb{E}_{t-\tau} \left\langle \frac{1}{N} \sum_i \frac{1}{K_{i,t}} \sum_{k=0}^{K_{i,t}-1} \bar{g}_i(\theta_i^{t,k}) - \bar{g}_i(\bar{\theta}^t), \bar{\theta}^t - \theta^* \right\rangle + \mathbb{E}_{t-\tau} \left\langle \frac{1}{N} \sum_i \frac{1}{K_{i,t}} \sum_{j=1}^{K_{i,t}} \bar{g}(\bar{\theta}^t), \bar{\theta}^t - \theta^* \right\rangle
$$

$$
\leq \frac{1}{N} \sum_i \frac{1}{K_{i,t}} \sum_{k=0}^{K_{i,t}-1} \left( \frac{1}{2\xi_2} \mathbb{E}_{t-\tau} \left\| \bar{g}_i(\theta_i^{t,k}) - \bar{g}_i(\bar{\theta}^t) \right\|_2^2 + \frac{\xi_2}{2} \mathbb{E}_{t-\tau} \left\| \bar{\theta}^t - \theta^* \right\|_2^2 \right)
$$
$$
+ \mathbb{E}_{t-\tau} \left\langle \bar{g}(\bar{\theta}^t), \bar{\theta}^t - \theta^* \right\rangle \tag{51}
$$

$$
\leq \frac{2}{\xi_2} \cdot \frac{1}{N} \sum_i \frac{1}{K_{i,t}} \sum_{k=0}^{K_{i,t}-1} \mathbb{E}_{t-\tau} \left\| \theta_i^{t,k} - \bar{\theta}^t \right\|_2^2 + \frac{\xi_2}{2} \mathbb{E}_{t-\tau} \left\| \bar{\theta}^t - \theta^* \right\|_2^2 + \mathbb{E}_{t-\tau} \left\langle \bar{g}(\bar{\theta}^t), \bar{\theta}^t - \theta^* \right\rangle \tag{52}
$$

where (51) follows from Young's inequality, $\xi_2$ is a positive number and (52) follows from Assumption 3.

For the third term $W_2$ in (50), we have:

$$
W_2 = \mathbb{E}_{t-\tau} \left\| \frac{1}{N} \sum_i (d_i^{(t)} - h_i^{(t)}) + h_i^{(t)} \right\|_2^2
$$

$$
\leq 2\, \mathbb{E}_{t-\tau} \underbrace{\left\| \frac{1}{N} \sum_i \frac{1}{K_{i,t}} \sum_{k=0}^{K_{i,t}-1} \left( A_i(O_i^{t,k}) - \bar{A}_i \right) (\theta_i^{t,k} - \theta_i^*) + Z_i(O_i^{t,k}) \right\|_2^2}_{W_{21}} + 2\, \mathbb{E}_{t-\tau} \underbrace{\left\| \frac{1}{N} \sum_i h_i^{(t)} \right\|_2^2}_{W_{22}}
$$
$$
\tag{53}
$$

Then we can bound $W_{21}$,

$$
W_{21} \leq 2\mathbb{E}_{t-\tau} \left\| \frac{1}{N} \sum_i \frac{1}{K_{i,t}} \sum_{k=0}^{K_{i,t}-1} \left( A_i(O_i^{t,k}) - \bar{A}_i \right) (\theta_i^{t,k} - \theta_i^*) \right\|_2^2
$$

$$
+ 2\mathbb{E}_{t-\tau} \left\| \frac{1}{N} \sum_i \frac{1}{K_{i,t}} \sum_{k=0}^{K_{i,t}-1} Z_i(O_i^{t,k}) \right\|_2^2
$$

$$
\leq 8c^2 \mathbb{E}_{t-\tau} \left\| \frac{1}{N} \sum_i \frac{1}{K_{i,t}} \sum_{k=0}^{K_{i,t}-1} (\theta_i^{t,k} - \bar{\theta}^t + \bar{\theta}^t - \theta_i^*) \right\|_2^2 + 2\mathbb{E}_{t-\tau} \left\| \frac{1}{N} \sum_i \frac{1}{K_{i,t}} \sum_{k=0}^{K_{i,t}-1} Z_i(O_i^{t,k}) \right\|_2^2
$$
$$
\tag{54}
$$

$$
\leq 16c^2 \mathbb{E}_{t-\tau} \Omega_t + \frac{16c^2}{N} \mathbb{E}_{t-\tau} \sum_i \left\| \bar{\theta}^t - \theta_i^* \right\|_2^2 + 2\mathbb{E}_{t-\tau} \left\| \frac{1}{N} \sum_i \frac{1}{K_{i,t}} \sum_{k=0}^{K_{i,t}-1} Z_i(O_i^{t,k}) \right\|_2^2
$$

$$
\leq 16c^2 \mathbb{E}_{t-\tau} \Omega_t + 16c^2 H^2 + 2\mathbb{E}_{t-\tau} \left\| \frac{1}{N} \sum_i \frac{1}{K_{i,t}} \sum_{k=0}^{K_{i,t}-1} Z_i(O_i^{t,k}) \right\|_2^2 \tag{55}
$$

where $\Omega_t = \frac{1}{N} \sum_i \frac{1}{K_{i,t}} \sum_{k=0}^{K_{i,t}-1} \left\| \theta_i^{t,k} - \bar{\theta}^t \right\|_2^2$, and (54) follows from Lemma 4.

Then for $W_{22}$,

$$W_{22} = \mathbb{E}_{t-\tau} \left\| \frac{1}{N} \sum_i \frac{1}{K_{i,t}} \sum_{k=0}^{K_{i,t}-1} \left( \bar{g}_i(\theta_i^{t,k}) - \bar{g}_i(\bar{\theta}^t) + \bar{g}_i(\bar{\theta}^t) - \bar{g}(\bar{\theta}^t) - \bar{g}(\bar{\theta}^t) \right) \right\|_2^2$$

$$\leq 2\mathbb{E}_{t-\tau} \left\| \frac{1}{N} \sum_i \frac{1}{K_{i,t}} \sum_{k=0}^{K_{i,t}-1} \bar{g}_i(\theta_i^{t,k}) - \bar{g}_i(\bar{\theta}^t) \right\|_2^2 + 2\mathbb{E}_{t-\tau} \left\| \bar{g}(\bar{\theta}^t) \right\|_2^2 \tag{56}$$

$$\leq \frac{2}{N} \sum_i \frac{1}{K_{i,t}} \sum_{k=0}^{K_{i,t}-1} \mathbb{E}_{t-\tau} \left\| \bar{g}_i(\theta_i^{t,k}) - \bar{g}_i(\bar{\theta}^t) \right\|_2^2 + 2\mathbb{E}_{t-\tau} \left\| \bar{g}(\bar{\theta}^t) \right\|_2^2$$

$$\leq 8\Omega_t + 2\mathbb{E}_{t-\tau} \left\| \bar{g}(\bar{\theta}^t) \right\|_2^2 \tag{57}$$

where (56) follows from the fact that $\sum_i \left\| \frac{1}{N} z_i \right\|^2 \leq \frac{1}{N} \sum_i \left\| z_i \right\|^2$, and (57) follows from Assumption 3.

By combining the terms $W_{21}$ and $W_{22}$, we can get

$$W_2 \leq 16(2c^2+1)\Omega_t + 32c^2 \left( \beta^2 \left\| \bar{\theta}^t - \theta^* \right\|_2^2 + \kappa^2 \right)$$

$$+ 4\mathbb{E}_{t-\tau} \left\| \frac{1}{N} \sum_i \frac{1}{K_{i,t}} \sum_{k=0}^{K_{i,t}-1} Z_i(O_i^{t,k}) \right\|_2^2 + 4\mathbb{E}_{t-\tau} \left\| \bar{g}(\bar{\theta}^t) \right\|_2^2 \tag{58}$$

Next, we study the gradient bias term $W_3$ which plays a central role in Markovian setting. Similar to the analysis of this bias due to Markovian noise in Khodadadian et al. (2022); Wang et al. (2023), we rewrite it as

$$W_3 = \mathbb{E}_{t-\tau} \left\langle \frac{1}{N} \sum_i d_i^{(t)} - h_i^{(t)}, \bar{\theta}^t - \theta^* \right\rangle$$

$$= \underbrace{\mathbb{E}_{t-\tau} \left\langle \frac{1}{N} \sum_i \frac{1}{K_{i,t}} \sum_{k=0}^{K_{i,t}-1} g_i(\theta_i^{t,k}) - \bar{g}_i(\theta_i^{t,k}), \bar{\theta}^t - \bar{\theta}^{t-\tau} \right\rangle}_{W_{31}}$$

$$+ \underbrace{\mathbb{E}_{t-\tau} \left\langle \frac{1}{N} \sum_i \frac{1}{K_{i,t}} \sum_{k=0}^{K_{i,t}-1} g_i(\theta_i^{t,k}) - g_i(\theta_i^{t-\tau,k}) + \bar{g}_i(\theta_i^{t-\tau,k}) - \bar{g}_i(\theta_i^{t,k}), \bar{\theta}^{t-\tau} - \theta^* \right\rangle}_{W_{32}}$$

$$+ \underbrace{\mathbb{E}_{t-\tau} \left\langle \frac{1}{N} \sum_i \frac{1}{K_{i,t}} \sum_{k=0}^{K_{i,t}-1} g_i(\theta_i^{t-\tau,k}) - \bar{g}_i(\theta_i^{t-\tau,k}), \bar{\theta}^{t-\tau} - \theta^* \right\rangle}_{W_{33}}$$

For the term $W_{31}$, we can conclude that

$$W_{31} = \mathbb{E}_{t-\tau} \left\langle \frac{1}{N} \sum_i \frac{1}{K_{i,t}} \sum_{k=0}^{K_{i,t}-1} g_i(\theta_i^{t,k}) - \bar{g}_i(\theta_i^{t,k}), \bar{\theta}^t - \bar{\theta}^{t-\tau} \right\rangle$$

$$\leq \mathbb{E}_{t-\tau} \left[ \left\| \frac{1}{N} \sum_i \frac{1}{K_{i,t}} \sum_{k=0}^{K_{i,t}-1} g_i(\theta_i^{t,k}) - \bar{g}_i(\theta_i^{t,k}) \right\|_2 \cdot \left\| \bar{\theta}^t - \bar{\theta}^{t-\tau} \right\|_2 \right]$$

$$\leq \mathbb{E}_{t-\tau} \left[ \frac{\xi_3}{2} \left\| \frac{1}{N} \sum_i \frac{1}{K_{i,t}} \sum_{k=0}^{K_{i,t}-1} g_i(\theta_i^{t,k}) - \bar{g}_i(\theta_i^{t,k}) \right\|_2^2 + \frac{1}{2\xi_3} \left\| \bar{\theta}^t - \bar{\theta}^{t-\tau} \right\|_2^2 \right]$$

where $\xi_3$ can be any real positive number. From (55), we have:

$$W_{31} \leq 8c^2 \xi_3 \Omega_t + 8\beta^2 c^2 \xi_3 \mathbb{E}_{t-\tau} \left\| \bar{\theta}^t - \theta^* \right\|_2^2 + 8c^2 \kappa^2 \xi_3$$

$$+ \xi_3 \mathbb{E}_{t-\tau} \left\| \frac{1}{N} \sum_i \frac{1}{K_{i,t}} \sum_{k=0}^{K_{i,t}-1} Z_i(O_i^{t,k}) \right\|_2^2 + \frac{1}{2\xi_3} \left\| \bar{\theta}^t - \bar{\theta}^{t-\tau} \right\|_2^2$$

For the term $W_{32}$, we have

$$W_{32} = \mathbb{E}_{t-\tau} \left\langle \frac{1}{N} \sum_i \frac{1}{K_{i,t}} \sum_{k=0}^{K_{i,t}-1} g_i(\theta_i^{t,k}) - g_i(\theta_i^{t-\tau,k}) + \bar{g}_i(\theta_i^{t-\tau,k}) - \bar{g}_i(\theta_i^{t,k}), \bar{\theta}^{t-\tau} - \theta^* \right\rangle$$

$$\leq \mathbb{E}_{t-\tau} \left\| \frac{1}{N} \sum_i \frac{1}{K_{i,t}} \sum_{k=0}^{K_{i,t}-1} g_i(\theta_i^{t,k}) - g_i(\theta_i^{t-\tau,k}) + \bar{g}_i(\theta_i^{t-\tau,k}) - \bar{g}_i(\theta_i^{t,k}) \right\|_2 \left\| \bar{\theta}^{t-\tau} - \theta^* \right\|_2$$

$$\leq \mathbb{E}_{t-\tau} \left[ \frac{1}{N} \sum_i \frac{1}{K_{i,t}} \sum_{k=0}^{K_{i,t}-1} \left\| g_i(\theta_i^{t,k}) - g_i(\theta_i^{t-\tau,k}) \right\|_2 + \left\| \bar{g}_i(\theta_i^{t-\tau,k}) - \bar{g}_i(\theta_i^{t,k}) \right\|_2 \right] \left\| \bar{\theta}^{t-\tau} - \theta^* \right\|_2$$

$$\leq 4 \mathbb{E}_{t-\tau} \left[ \frac{1}{N} \sum_i \frac{1}{K_{i,t}} \sum_{k=0}^{K_{i,t}-1} \left\| \theta_i^{t,k} - \theta_i^{t-\tau,k} \right\|_2 \right] \left\| \bar{\theta}^{t-\tau} - \theta^* \right\|_2 \tag{59}$$

$$\leq 4 \mathbb{E}_{t-\tau} \left[ \frac{1}{N} \sum_i \frac{1}{K_{i,t}} \sum_{k=0}^{K_{i,t}-1} \left\| \theta_i^{t,k} - \bar{\theta}^t \right\|_2 + \frac{1}{N} \sum_i \frac{1}{K_{i,t}} \sum_{k=0}^{K_{i,t}-1} \left\| \bar{\theta}^t - \bar{\theta}^{t-\tau} \right\|_2 \right.$$

$$+ \frac{1}{N} \sum_i \frac{1}{K_{i,t}} \sum_{k=0}^{K_{i,t}-1} \left\| \bar{\theta}^{t-\tau} - \theta_i^{t-\tau,k} \right\|_2 \right] \left[ \left\| \bar{\theta}^{t-\tau} - \bar{\theta}^t \right\|_2 + \left\| \bar{\theta}^t - \theta^* \right\|_2 \right]$$

$$= 4 \mathbb{E}_{t-\tau} \left[ \Delta_t \left\| \bar{\theta}^{t-\tau} - \bar{\theta}^t \right\|_2 + \left\| \bar{\theta}^t - \bar{\theta}^{t-\tau} \right\|_2^2 + \Delta_{t-\tau} \left\| \bar{\theta}^{t-\tau} - \bar{\theta}^t \right\|_2 \right.$$

$$+ \Delta_t \left\| \bar{\theta}^t - \theta^* \right\|_2 + \left\| \bar{\theta}^t - \bar{\theta}^{t-\tau} \right\|_2 \left\| \bar{\theta}^t - \theta^* \right\|_2 + \Delta_{t-\tau} \left\| \bar{\theta}^t - \theta^* \right\|_2 \right]$$

$$\leq 2 \mathbb{E}_{t-\tau} \left[ \frac{2}{\xi_4} \Delta_t^2 + \frac{2}{\xi_4} \Delta_{t-\tau}^2 + (2\xi_4 + \frac{1}{\xi_4} + 2) \left\| \bar{\theta}^{t-\tau} - \bar{\theta}^t \right\|_2^2 + 3\xi_4 \left\| \bar{\theta}^t - \theta^* \right\|_2^2 \right] \tag{60}$$

$$\leq 2 \mathbb{E}_{t-\tau} \left[ \frac{2}{\xi_4} \Omega_t + \frac{2}{\xi_4} \Omega_{t-\tau} + (2\xi_4 + \frac{1}{\xi_4} + 2) \left\| \bar{\theta}^{t-\tau} - \bar{\theta}^t \right\|_2^2 + 3\xi_4 \left\| \bar{\theta}^t - \theta^* \right\|_2^2 \right] \tag{61}$$

where $\xi_4$ can be any positive real number and $\Delta_t = \frac{1}{N} \sum_i \frac{1}{K_{i,t}} \sum_{k=0}^{K_{i,t}-1} \left\| \theta_i^{t,k} - \bar{\theta}^t \right\|_2$; $\Omega_t = \sum_i \frac{1}{K_{i,t}} \sum_{k=0}^{K_{i,t}-1} \left\| \theta_i^{t,k} - \bar{\theta}^t \right\|_2^2$; (59) follows from the Lipschitz property by Assumption 3 and the triangle inequality; (60) follows from Young's inequality; (61) is derived from the convexity of a square function as follows.

$$\mathbb{E}_{t-\tau} \left\| \frac{1}{N} \sum_i \frac{1}{K_{i,t}} \sum_{k=0}^{K_{i,t}-1} \theta_i^{t,k} - \bar{\theta}^t \right\|_2^2 \leq \frac{1}{N} \sum_i \frac{1}{K_{i,t}} \sum_{k=0}^{K_{i,t}-1} \mathbb{E}_{t-\tau} \left\| \theta_i^{t,k} - \bar{\theta}^t \right\|_2^2 \tag{62}$$

For the third term of $W_{33}$, we have

$$W_{33} = \mathbb{E}_{t-\tau} \left\langle \frac{1}{N} \sum_i \frac{1}{K_{i,t}} \sum_{k=0}^{K_{i,t}-1} g_i(\theta_i^{t-\tau,k}) - \bar{g}_i(\theta_i^{t-\tau,k}), \bar{\theta}^{t-\tau} - \theta^* \right\rangle$$

$$= \frac{1}{N} \mathbb{E}_{t-\tau} \sum_i \frac{1}{K_{i,t}} \sum_{k=0}^{K_{i,t}-1} \left\| g_i(\theta_i^{t-\tau,k}) - \bar{g}_i(\theta_i^{t-\tau,k}) \right\|_2 \left\| \bar{\theta}^{t-\tau} - \theta^* \right\|_2.$$

Then we can further bound it by Lemma 5 as

$$
\begin{aligned}
W_{33} &\leq \frac{1}{N}\mathbb{E}_{t-\tau}\sum_i \frac{1}{K_{i,t}}\sum_{k=0}^{K_{i,t}-1}\left(\left\|(A_i(O_i^{t,k})-\bar{A}_i)(\theta_i^{t-\tau,k}-\theta_i^*)\right\|_2 + \left\|Z_i(O_i^{(t,k)})\right\|_2\right)\left\|\bar{\theta}^{t-\tau}-\theta^*\right\|_2 \\
&\leq \frac{1}{N}\mathbb{E}_{t-\tau}\sum_i \frac{1}{K_{i,t}}\sum_{k=0}^{K_{i,t}-1}\left(c'\rho^{\sum_{p=t-\tau}^{t-1}K_{i,p}+k}\left\|(\theta_i^{t-\tau,k}-\theta_i^*)\right\|_2 + q'\rho^{\sum_{p=t-\tau}^{t-1}K_{i,p}+k}\right)\left\|\bar{\theta}^{t-\tau}-\theta^*\right\|_2 \\
&\leq \frac{\alpha}{N}\mathbb{E}_{t-\tau}\sum_i \frac{1}{K_{i,t}}\sum_{k=0}^{K_{i,t}-1}\left(c'\left\|(\theta_i^{t-\tau,k}-\theta_i^*)\right\|_2 + q'\right)\left\|\bar{\theta}^{t-\tau}-\theta^*\right\|_2 \qquad (63) \\
&\leq \frac{\alpha}{N}\mathbb{E}_{t-\tau}\sum_i \frac{1}{K_{i,t}}\sum_{k=0}^{K_{i,t}-1}\left(c'\left\|\theta_i^{t-\tau,k}-\bar{\theta}^{t-\tau}\right\|_2 + c'\left\|\bar{\theta}^{t-\tau}-\bar{\theta}^t\right\|_2 + c'\left\|\bar{\theta}^t-\theta_i^*\right\|_2 + q'\right)H \\
&\leq \alpha\left[\frac{1}{2}c'\Omega_{t-\tau} + 2c'H^2 + \frac{1}{2}c'\left(\beta^2\mathbb{E}_{t-\tau}\left\|\bar{\theta}^t-\theta^*\right\|_2^2 + \kappa^2\right) + q'H\right]
\end{aligned}
$$

where (63) is derived from the fact that $\rho^{\sum_{p=t-\tau}^{t-1}K_{i,p}} \leq \alpha$ for all $i$.

## J.3 DRIFT TERM $\Omega_t$

For the drift term $\Omega_t$, we begin with

$$
\begin{aligned}
\mathbb{E}\left\|\theta_i^{t,k}-\bar{\theta}^t\right\|_2^2 &= \alpha^2\mathbb{E}\left\|\sum_{s=0}^{k-1}g_i(\theta_i^{t,s})\right\|_2^2 \\
&= \alpha^2\mathbb{E}\left\|\sum_{s=0}^{k-1}(A_i(O_i^{t,s})(\theta_i^{t,s}-\theta_i^*)+Z_i(O_i^{t,s}))\right\|_2^2 \\
&\leq 2\alpha^2\mathbb{E}\left\|\sum_{s=0}^{k-1}A_i(O_i^{t,s})(\theta_i^{t,s}-\theta_i^*)\right\|_2^2 + 2\alpha^2\mathbb{E}\left\|\sum_{s=0}^{k-1}Z_i(O_i^{t,s})\right\|_2^2 \\
&\leq 2\alpha^2c^2\mathbb{E}\left\|\sum_{s=0}^{k-1}(\theta_i^{t,s}-\theta_i^*)\right\|_2^2 + 2\alpha^2\mathbb{E}\left\|\sum_{s=0}^{k-1}Z_i(O_i^{t,s})\right\|_2^2 \qquad (64) \\
&\leq 2\alpha^2c^2k\sum_{s=0}^{k-1}\mathbb{E}\left\|\theta_i^{t,s}-\theta_i^*\right\|_2^2 + 2\alpha^2k^2q^2 \qquad (65) \\
&\leq 4\alpha^2c^2k\sum_{s=0}^{k-1}\mathbb{E}\left\|\theta_i^{t,s}-\bar{\theta}^t\right\|_2^2 + 4\alpha^2c^2k^2\mathbb{E}\left\|\bar{\theta}^t-\theta_i^*\right\|_2^2 + 2\alpha^2k^2q^2
\end{aligned}
$$

where (64) and (65) are derived from Lemma 4. Similar to the analysis in the I.I.D. setting, we have

$$
\begin{aligned}
\frac{1}{K_{i,t}}\sum_{k=0}^{K_{i,t}-1}\mathbb{E}\left\|\theta_i^{t,k}-\bar{\theta}^t\right\|_2^2 &\leq 2\alpha^2c^2(K_{i,t}-1)\sum_{k=0}^{K_{i,t}-1}\mathbb{E}\left\|\theta_i^{t,k}-\bar{\theta}^t\right\|_2^2 \\
&\quad + \frac{2\alpha^2c^2(K_{i,t}-1)(2K_{i,t}-1)}{3}\mathbb{E}\left\|\bar{\theta}^t-\theta_i^*\right\|_2^2 \\
&\quad + \frac{\alpha^2q^2(K_{i,t}-1)(2K_{i,t}-1)}{3}
\end{aligned}
$$

After minor arrangements, we have

$$\frac{1}{K_{i,t}} \sum_{k=0}^{K_{i,t}-1} \mathbb{E} \left\| \theta_i^{t,k} - \bar{\theta}^t \right\|_2^2 \leq \frac{2\alpha^2 c^2 (K_{i,t}-1)(2K_{i,t}-1)}{3(1-2\alpha^2 c^2 K_{i,t}(K_{i,t}-1))} \mathbb{E} \left\| \bar{\theta}^t - \theta_i^* \right\|_2^2$$

$$+ \frac{\alpha^2 q^2 (K_{i,t}-1)(2K_{i,t}-1)}{3(1-2\alpha^2 c^2 K_{i,t}(K_{i,t}-1))}$$

Define $C' = \max_i \{2\alpha^2 c^2 K_{i,t}(K_{i,t}-1)\}$, then we have

$$\frac{1}{K_{i,t}} \sum_{k=0}^{K_{i,t}-1} \mathbb{E} \left\| \theta_i^{t,k} - \bar{\theta}^t \right\|_2^2 \leq \frac{2\alpha^2 c^2 (K_{i,t}-1)(2K_{i,t}-1)}{3(1-C')} \mathbb{E} \left\| \bar{\theta}^t - \theta_i^* \right\|_2^2$$

$$+ \frac{\alpha^2 q^2 (K_{i,t}-1)(2K_{i,t}-1)}{3(1-C')} \tag{66}$$

Then, the drift term can be represented as

$$\frac{1}{N} \sum_i \frac{1}{K_{i,t}} \sum_{k=0}^{K_{i,t}-1} \mathbb{E} \left\| \theta_i^{t,k} - \bar{\theta}^t \right\|_2^2$$

$$\leq \frac{2\alpha^2 c^2 (K_{t,\max}-1)(2K_{t,\max}-1)}{3(1-C')N} \sum_i \mathbb{E} \left\| \bar{\theta}^t - \theta_i^* \right\|_2^2 + \frac{\alpha^2 q^2 (K_{t,\max}-1)(2K_{t,\max}-1)}{3(1-C')}$$

$$\leq \frac{2\alpha^2 c^2 (K_{t,\max}-1)(2K_{t,\max}-1)}{3(1-C')} \left( \beta^2 \mathbb{E} \left\| \bar{\theta}^t - \theta^* \right\|_2^2 + \kappa^2 \right)$$

$$+ \frac{\alpha^2 q^2 (K_{t,\max}-1)(2K_{t,\max}-1)}{3(1-C')} \tag{67}$$

Next, we bound $\mathbb{E} \left\| \bar{\theta}^t - \bar{\theta}^{t-\tau} \right\|^2$ since it appears in $W_3$.

$$\left\| \bar{\theta}^{l+1} - \bar{\theta}^l \right\|_2^2 = \alpha^2 \left\| \frac{1}{N} \sum_i \frac{1}{K_{i,t}} \sum_{k=0}^{K_{i,t}-1} g_i(\theta_i^{l,k}) \right\|_2^2$$

$$= \alpha^2 \left\| \frac{1}{N} \sum_i \frac{1}{K_{i,t}} \sum_{k=0}^{K_{i,t}-1} (A_i(O_i^{l,k})(\theta_i^{l,k} - \theta_i^*) + Z_i(O_i^{l,k})) \right\|_2^2$$

$$\leq 2c\alpha^2 \left\| \frac{1}{N} \sum_i \frac{1}{K_{i,t}} \sum_{k=0}^{K_{i,t}-1} \theta_i^{l,k} - \theta_i^* \right\|_2^2 + 2\alpha^2 \left\| \frac{1}{N} \sum_i \frac{1}{K_{i,t}} \sum_{k=0}^{K_{i,t}-1} Z_i(O_i^{l,k}) \right\|_2^2$$

$$\leq 4c\alpha^2 \left\| \frac{1}{N} \sum_i \frac{1}{K_{i,t}} \sum_{k=0}^{K_{i,t}-1} (\theta_i^{l,k} - \bar{\theta}^t) \right\|_2^2 + 4c\alpha^2 \left\| \frac{1}{N} \sum_i \frac{1}{K_{i,t}} \sum_{k=0}^{K_{i,t}-1} (\bar{\theta}^t - \theta_i^*) \right\|_2^2$$

$$+ 2\alpha^2 \left\| \frac{1}{N} \sum_i \frac{1}{K_{i,t}} \sum_{k=0}^{K_{i,t}-1} Z_i(O_i^{l,k}) \right\|_2^2$$

$$\leq 4c\alpha^2 \left\| \frac{1}{N} \sum_i \frac{1}{K_{i,t}} \sum_{k=0}^{K_{i,t}-1} (\theta_i^{l,k} - \bar{\theta}^t) \right\|_2^2 + 4c\alpha^2 \left( \beta^2 \left\| \bar{\theta}^t - \theta^* \right\|_2^2 + \kappa^2 \right)$$

$$+ 2\alpha^2 \left\| \frac{1}{N} \sum_i \frac{1}{K_{i,t}} \sum_{k=0}^{K_{i,t}-1} Z_i(O_i^{l,k}) \right\|_2^2 \tag{68}$$

Then we can establish the bound in conditional expectation on $t - 2\tau$:

$$\mathbb{E}_{t-2\tau} \left\| \bar{\theta}^t - \bar{\theta}^{t-\tau} \right\|_2^2$$

$$\leq \tau \sum_{s=t-\tau}^{t-1} \left\| \bar{\theta}^{s+1} - \bar{\theta}^s \right\|_2^2$$

$$\leq 2\alpha^2 \tau \sum_{s=t-\tau}^{t-1} \mathbb{E}_{t-2\tau} \left[ 2c \left\| \frac{1}{N} \sum_i \frac{1}{K_{i,t}} \sum_{k=0}^{K_{i,t}-1} (\theta_i^{s,k} - \bar{\theta}^t) \right\|_2^2 + 2c \left( \beta^2 \left\| \bar{\theta}^s - \theta^* \right\|_2^2 + \kappa^2 \right) \right.$$

$$\left. + \left\| \frac{1}{N} \sum_i \frac{1}{K_{i,t}} \sum_{k=0}^{K_{i,t}-1} Z_i(O_i^{s,k}) \right\|_2^2 \right] \tag{69}$$

$$\leq 4\alpha^2 c\tau \sum_{s=0}^{\tau} \mathbb{E}_{t-2\tau} \Omega_{t-s} + 4\alpha^2 c\tau^2 \left( \beta^2 H^2 + \kappa^2 \right)$$

$$+ 2\alpha^2 \tau^2 \mathbb{E}_{t-2\tau} \left[ \frac{1}{N^2} \sum_i \frac{[q']^2}{K_{i,t}} + \frac{2qq'\rho(K_{t,\max} - 1)}{N^2(1-\rho)} \sum_i \frac{1}{K_{i,t}^2} \right.$$

$$\left. + \frac{2[q']^2 \rho(K_{t,\max} - 1)}{N^2(1-\rho)} \sum_{i<j} \frac{1}{K_{i,t} K_{j,t}} \rho^{\sum_{p=t-2\tau}^{t-1}(K_{i,p} + K_{j,p})} \right] \tag{70}$$

where (69) follows from (68); (70) follows from Lemma 4; (146) follows from that fact that $\rho^{K_{i,t}\tau} \leq \rho^{K_{t,\min}\tau} \leq \alpha$ for all $t$.

Finally, we choose $\tau = \left\lfloor \frac{\log_\rho \alpha}{K_{t,\min}} \right\rfloor$ and by combining $W_1$, $W_2$ and $W_3$, we have:

$$\mathbb{E}_{t-2\tau} \left\| \bar{\theta}^{t+1} - \theta^* \right\|_2^2$$

$$\leq \mathbb{E}_{t-2\tau} \left\| \bar{\theta}^t - \theta^* \right\|_2^2 + 2\alpha_t \bar{K}_t \mathbb{E}_{t-2\tau} \left\langle \frac{1}{N} \sum_i h_i^{(t)}, \bar{\theta}^t - \theta^* \right\rangle + \alpha_t^2 \bar{K}_t^2 \mathbb{E}_{t-2\tau} \left\| \frac{1}{N} \sum_i d_i^{(t)} \right\|_2^2$$

$$+ 2\alpha_t \bar{K}_t \mathbb{E}_{t-2\tau} \left\langle \frac{1}{N} \sum_i (d_i^{(t)} - h_i^{(t)}), \bar{\theta}^t - \theta^* \right\rangle$$

$$\leq 2\alpha_t \bar{K}_t \left( \frac{2}{\xi_2} \cdot \mathbb{E}_{t-2\tau} \Omega_t + \frac{\xi_2}{2} \mathbb{E}_{t-2\tau} \left\| \bar{\theta}^t - \theta^* \right\|_2^2 + \mathbb{E}_{t-2\tau} \left\langle \bar{g}(\bar{\theta}^t), \bar{\theta}^t - \theta^* \right\rangle \right)$$

$$+ 16\alpha_t^2 \bar{K}_t^2 \left( (2c^2 + 1)\mathbb{E}_{t-2\tau} \Omega_t + 2c^2 \left( \beta^2 \mathbb{E} \left\| \bar{\theta}^t - \theta^* \right\|_2^2 + \kappa^2 \right) \right)$$

$$+ 4\alpha_t^2 \bar{K}_t^2 \left( \mathbb{E}_{t-2\tau} \left\| \frac{1}{N} \sum_i \frac{1}{K_{i,t}} \sum_{k=0}^{K_{i,t}-1} Z_i(O_i^{t,k}) \right\|_2^2 + \mathbb{E}_{t-2\tau} \left\| \bar{g}(\bar{\theta}^t) \right\|_2^2 \right)$$

$$+ 2\alpha_t \bar{K}_t \left( 8c^2 \xi_3 \mathbb{E}_{t-2\tau} \Omega_t + 8\beta^2 c^2 \xi_3 \mathbb{E}_{t-2\tau} \left\| \bar{\theta}^t - \theta^* \right\|_2^2 + 8c^2 \kappa^2 \xi_3 \right)$$

$$+ 2\alpha_t \bar{K}_t \left( \xi_3 \mathbb{E}_{t-2\tau} \left\| \frac{1}{N} \sum_i \frac{1}{K_{i,t}} \sum_{k=0}^{K_{i,t}-1} Z_i(O_i^{t,k}) \right\|_2^2 + \frac{1}{2\xi_3} \mathbb{E}_{t-2\tau} \left\| \bar{\theta}^t - \bar{\theta}^{t-\tau} \right\|_2^2 \right)$$

$$+ 4\alpha_t \bar{K}_t \mathbb{E}_{t-2\tau} \left[ \frac{2}{\xi_4} \Omega_t + \frac{2}{\xi_4} \Omega_{t-\tau} + \left( 2\xi_4 + \frac{1}{\xi_4} + 2 \right) \left\| \bar{\theta}^{t-\tau} - \bar{\theta}^t \right\|_2^2 + 3\xi_4 \left\| \bar{\theta}^t - \theta^* \right\|_2^2 \right]$$

$$+ 2\alpha_t^2 \bar{K}_t \left( \frac{1}{2} c' \mathbb{E}_{t-2\tau} \Omega_{t-\tau} + 3c' H^2 + q' H \right) \tag{71}$$

where (71) is derived from (52), (58), (61), and (55).

If we choose $\xi_2 = \frac{\lambda}{4}$, $\xi_3 = \alpha_t$, $\xi_4 = \frac{\lambda}{48}$, and step size $\alpha_t \leq \frac{\lambda}{4\left(32\bar{K}_t^2\beta^2 c^2 + 16\bar{K}_t^2 + 16\beta^2 c^2 + \beta^2 c'\right)}$, by combining (67) and Lemma 4, we have

$$
\mathbb{E}_{t-2\tau}\left\|\bar{\theta}^{t+1} - \theta^*\right\|_2^2 \leq \left(1 - \frac{\alpha\bar{K}_t\lambda}{4}\right)\mathbb{E}_{t-2\tau}\left\|\bar{\theta}^t - \theta^*\right\|_2^2
$$
$$
+ \left[\frac{400\alpha_t\bar{K}_t}{\lambda} + 16\alpha_t^2\bar{K}_t^2(2c^2+1) + 16\alpha_t^2\bar{K}_t c^2\right]\mathbb{E}_{t-2\tau}\Omega_t
$$
$$
+ \left(\frac{384\alpha_t\bar{K}_t}{\lambda} + \alpha_t^2\bar{K}_t c'\right)\mathbb{E}_{t-2\tau}\Omega_{t-\tau} + \left[\bar{K}_t + 4\alpha_t\bar{K}_t\left(\frac{\lambda}{24} + \frac{48}{\lambda} + 2\right)\right]\mathbb{E}_{t-2\tau}\left\|\bar{\theta}^t - \bar{\theta}^{t-\tau}\right\|_2^2
$$
$$
+ \left(4\alpha_t^2\bar{K}_t^2 + 2\alpha_t^2\bar{K}_t\right)\mathbb{E}_{t-2\tau}\left\|\frac{1}{N}\sum_i \frac{1}{K_{i,t}}\sum_{k=0}^{K_{i,t}-1}Z_i(O_i^{t,k})\right\|_2^2
$$
$$
+ \left(32\alpha_t^2\bar{K}_t^2 c^2 + \alpha_t^2\bar{K}_t c' + 16\alpha_t^2\bar{K}_t^2 c\right)\kappa^2 + 2\alpha_t^2\bar{K}_t H\left(2c'H + q'\right)
$$

If step sizes satisfy that

$$
\alpha_t \leq \min\left\{\frac{2}{\lambda(2c^2\bar{K}_t + c^2 + \bar{K}_t)}, \frac{1}{\lambda c'}, \frac{1}{4\left(\frac{\lambda}{24} + \frac{48}{\lambda} + 2\right)}\right\},
$$

then $\frac{400\alpha_t\bar{K}_t}{\lambda} + 16\alpha_t^2\bar{K}_t^2(2c^2+1) + 16\alpha_t^2\bar{K}_t c^2 \leq \frac{432\alpha_t\bar{K}_t}{\lambda}$, $\frac{384\alpha_t\bar{K}_t}{\lambda} + \alpha_t^2\bar{K}_t c' \leq \frac{385\alpha_t\bar{K}_t}{\lambda}$, and $\bar{K}_t + 4\alpha_t\bar{K}_t\left(\frac{\lambda}{24} + \frac{48}{\lambda} + 2\right) \leq 2\bar{K}_t$ hold. Then (71) can be simplified as

$$
\mathbb{E}_{t-2\tau}\left\|\bar{\theta}^{t+1} - \theta^*\right\|_2^2 \leq \left(1 - \frac{\alpha_t\bar{K}_t\lambda}{4}\right)\mathbb{E}_{t-2\tau}\left\|\bar{\theta}^t - \theta^*\right\|_2^2
$$
$$
+ \left(\frac{817\alpha_t\bar{K}_t}{\lambda} + 8\alpha_t^2\tau^2\bar{K}_t c\right)\left(\frac{2\alpha^2 c^2(K_{t,\max}-1)(2K_{t,\max}-1)}{3(1-C')}\left(\beta^2 H^2 + \kappa^2\right)\right)
$$
$$
+ \left(\frac{817\alpha_t\bar{K}_t}{\lambda} + 8\alpha_t^2\tau^2\bar{K}_t c\right)\left(\frac{\alpha^2 q^2(K_{t,\max}-1)(2K_{t,\max}-1)}{3(1-C')}\right)
$$
$$
+ \left(4\alpha_t^2\bar{K}_t^2 + 2\alpha_t^2\bar{K}_t + 4\alpha_t^2\tau^2\bar{K}_t\right)\left(\frac{1}{N^2}\sum_i \frac{[q']^2}{K_{i,t}} + \frac{2qq'\rho(K_{t,\max}-1)}{N^2(1-\rho)}\sum_i \frac{1}{K_{i,t}^2}\right)
$$
$$
+ \left(4\alpha_t^2\bar{K}_t^2 + 2\alpha_t^2\bar{K}_t + 4\alpha_t^2\tau^2\bar{K}_t\right)\left(\frac{2[c']^2\rho(K_{t,\max}-1)}{N^2(1-\rho)}\sum_{i<j}\frac{1}{K_{i,t}K_{j,t}}\rho^{\sum_{p=t-2\tau}^{t-1}(K_{i,p}+K_{j,p})}\right)
$$
$$
+ \left(32\alpha_t^2\bar{K}_t^2 c^2 + \alpha_t^2\bar{K}_t c' + 16\alpha_t^2\bar{K}_t^2 c + 8\alpha_t^2\tau^2\bar{K}_t c\right)\kappa^2 + 2\alpha_t^2\bar{K}_t H\left(2c'H + q' + 4\beta^2\tau^2 cH\right)
$$

Then let $K_{\max} = \max_{i,t}\{K_{i,t}\}$, $\bar{K}_{\max} = \max_t\{\bar{K}_t\}$ and $\bar{K}_{\min} = \min_t\{\bar{K}_t\}$. If $\alpha \leq \min_t\{\alpha_t\}$, applying the last inequality recursively, we have:

$$
\mathbb{E}\left\|\bar{\theta}^T - \theta^*\right\|_2^2 \leq e^{-\frac{\lambda\alpha\bar{K}_{\min}T}{4}}\mathbb{E}\left\|\bar{\theta}^0 - \theta^*\right\|_2^2
$$
$$
+ \left(\frac{817}{\lambda^2} + \frac{8\tau^2\alpha c}{\lambda}\right)\left(\frac{8\alpha^2 c^2(K_{\max}-1)(2K_{\max}-1)}{3(1-C')}\left(\beta^2 H^2 + \kappa^2\right)\right)
$$
$$
+ \left(\frac{817}{\lambda^2} + \frac{8\tau^2\alpha c}{\lambda}\right)\left(\frac{\alpha^2 q^2(K_{\max}-1)(2K_{\max}-1)}{3(1-C')}\right)
$$
$$
+ \frac{8\alpha\left(2\bar{K}_{\max} + 2\tau^2 + 1\right)}{\lambda}\left(\frac{[q']^2\hat{K}_{\max}}{N} + \frac{2qq'\rho(K_{\max}-1)\hat{K}_{\max}^2}{N(1-\rho)}\right)
$$
$$
+ \frac{8\alpha\left(2\bar{K}_{\max} + 2\tau^2 + 1\right)}{\lambda}\left(\frac{2[c']^2\rho(K_{\max}-1)}{N^2(1-\rho)}\sum_{i<j}\frac{1}{K_{i,t}K_{j,t}}\rho^{\sum_{s=0}^{2\tau-1}(K_{i,s}+K_{j,s})}\right)
$$
$$
+ \frac{4\alpha\left(32c^2\bar{K}_{\max} + 16\bar{K}_{\max}c + 8\tau^2 c + c'\right)\kappa^2 + 2\alpha H\left(2c'H + q' + 4\beta^2\tau^2 cH\right)}{\lambda} \tag{72}
$$

where $\hat{K}_t = \frac{1}{N} \sum_i \frac{1}{K_{i,t}}$, $\hat{K}_{\max} = \max_t \{\hat{K}_t\}$, $\hat{K}_t^2 = \frac{1}{N} \sum_i \frac{1}{K_{i,t}^2}$, and $\hat{K}_{\max}^2 = \max_t \{\hat{K}_t^2\}$.

## K   PROOF OF COROLLARY 2

Assuming we use a constant local update number $K$ for each agent, (72) can be simplified as

$$\mathbb{E}\left\|\bar{\theta}^T - \theta^*\right\|_2^2 \leq e^{-\frac{\lambda \alpha K T}{4}} \left\|\bar{\theta}^0 - \theta^*\right\|_2^2 \tag{73}$$

$$+ \left(\frac{817}{\lambda^2} + \frac{8\tau^2 \alpha c}{\lambda}\right) \left(\frac{8\alpha^2 c^2 (K-1)(2K-1)}{3(1-C')} \left(\beta^2 H^2 + \kappa^2\right)\right)$$

$$+ \left(\frac{817}{\lambda^2} + \frac{8\tau^2 \alpha c}{\lambda}\right) \left(\frac{\alpha^2 q^2 (K-1)(2K-1)}{3(1-C')}\right)$$

$$+ \frac{8\alpha \left(2K + 2\tau^2 + 1\right)}{\lambda} \left(\frac{[q']^2}{NK} + \frac{2qq'\rho(K-1)}{NK^2(1-\rho)}\right)$$

$$+ \frac{8\alpha \left(2K + 2\tau^2 + 1\right)}{\lambda} \left(\frac{[c']^2 (N-1)(K-1)\rho^{4K\tau+1}}{NK(1-\rho)}\right)$$

$$+ \frac{4\alpha \left(32c^2 K + 16Kc + 8\tau^2 c + c'\right)\kappa^2 + 2\alpha H \left(2c'H + q' + 4\beta^2 \tau^2 cH\right)}{\lambda} \tag{74}$$

If $N = 1$ and $K = 1$, then we have:

$$\mathbb{E}\left\|\bar{\theta}^T - \theta^*\right\|_2^2 \leq e^{-\frac{\lambda \alpha T}{4}} \left\|\bar{\theta}^0 - \theta^*\right\|_2^2 + \frac{2\alpha H \left(2c'H + q' + 4\beta^2 \tau^2 cH\right)}{\lambda} + \frac{8\alpha \left(3 + 2\tau^2\right)[q']^2}{\lambda}, \tag{75}$$

which matches the results of centralized TD as Theorem 3(b) in Bhandari et al. (2018).

## L    ADDITIONAL SIMULATIONS

In the simulations, the agent is initially placed in one corner of the maze and selects an action to move to the next cell with a certain probability. In the policy evaluation process, in order to avoid low learning efficiencies due to sparse rewards, the agent will receive a reward 0 if it reaches the desired goal and $-\frac{1}{2}\left[(x-3)^2+(y-3)^2\right]$ otherwise where $(x,y)$ is the position of the agent and it is also the current state. Here the state space size is 16 and the action can be selected from up, down, left, right directions. The goal of the agents is to learn a common model to approximate the value function under the given policy.

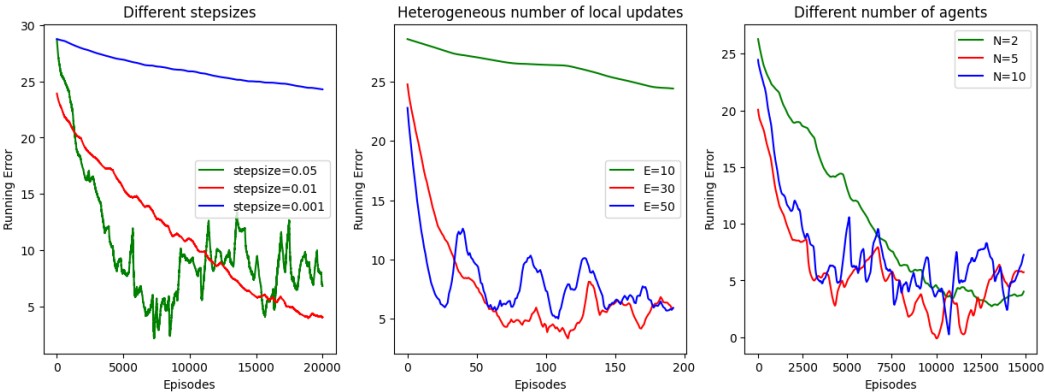

Figure 2: Training performance of HFTD with different settings under Markovian sampling: training performance refers to the running error between current model and optimal model. **Left:** agents perform local updates with different step sizes; **Middle:** agents take different number of local iterations with mean values 10, 30, and 50; **Right:** the number of agents participating the training is different.

Compared with the simulation results of I.I.D. sampling, the learning process under Markovian sampling shows more instability. This is because in Markovian sampling, the next state depends only on the current state under a fixed policy so that some states are not visited enough. Hence, it is difficult to approximate the value function well for the entire state space.

### L.1    MOUNTAIN CAR

For the mountain car experiment, we first discretize the continuous state spaces with a uniform grid by tile coding. Each tiling has $10 \times 10$ grid. The action can be selected from push left, no push, and push right. The goal of the agents is to learn a common model to approximate the value function under the given policy.

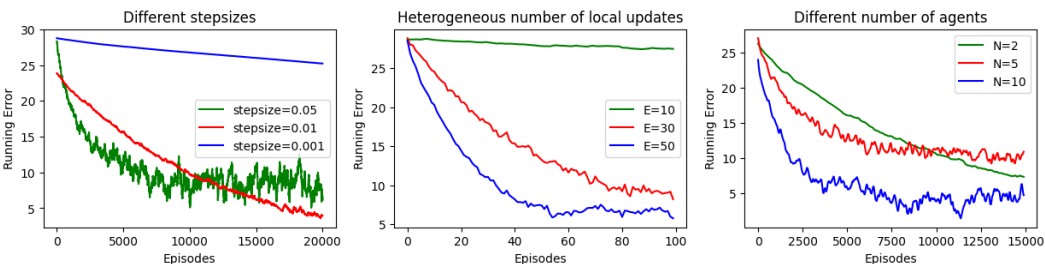

Figure 3: Training performance of HFTD with different settings under I.I.D sampling: training performance refers to the running error between current model and optimal model. **Left:** agents perform local updates with different step sizes; **Middle:** agents take different number of local iterations with mean values 10, 30, and 50; **Right:** the number of agents participating the training is different.

The simulation results show that the running error is larger than that of I.I.D. sampling in Gridworld with same heterogeneity level $\kappa^2 = 1$ . This is because in this setting, there exists discretization error which impedes the convergence performance. However, the simulation results show that the running error decreases with linear rate which matches our theoretical results. Furthermore, smaller step size lead to a less convergence error.