# OpenReview forum: "Finite-Time Analysis of Federated Temporal Difference Learning with Linear Function Approximation under Environment and Computation Heterogeneity"
_ICLR.cc/2024/Conference — Submitted to ICLR 2024_

### Official Review · Reviewer_Aqmo · 2023-10-31

**Soundness:** 4 excellent
**Presentation:** 4 excellent
**Contribution:** 3 good
**Rating:** 6
**Confidence:** 3

**Summary:**

This paper addresses the problem of federated reinforcement learning in the presence of environmental heterogeneity, where agents operate in diverse environments characterized by the same state and action spaces but distinct transition dynamics and reward functions. The authors derive perturbation bounds on the TD fixed points, which quantify the deviation of the fixed points based on the heterogeneity in the agents' MDPs. To establish the analysis results, the authors explore the properties of the virtual MDP, a concept introduced by the existing work on federated reinforcement learning with environmental heterogeneity. The paper's contributions include the development of a federated version of the TD algorithm tailored for this particular setup and the analysis results suggesting that under the assumption that all specified conditions are met, the study reveals that the linear convergence speedups are achievable with linearly parameterized models in a low-heterogeneity regime, which corresponds to scenarios approaching the homogenous case.

**Strengths:**

This paper offers a well-written and comprehensible exploration of the open challenge of improving the sample efficiency of RL agents by employing federated learning techniques with agents operating in heterogeneous environments. One of the key contributions of this paper is the investigation of the approximation error in federated learning as a function of environmental heterogeneity, which is then used to develop a federated version of the TD algorithm specifically tailored for the considered scenario. This algorithm leverages the advantages of federated learning to facilitate knowledge exchange and collaboration among the agents, improving the overall learning process. The result is significant, although I did not assess the correctness of all the proofs.

**Weaknesses:**

1. The simulations were conducted on the platform: GridWorld. Expanding simulations to more complex or diverse environments could provide a more comprehensive understanding of the algorithm's performance in practice.
2. While the authors verified theoretical results in a small-scale problem in a tabular form, it might be valuable to test on larger-scale problems or non-tabular formulations.
3. A deeper exploration of the algorithm's limitations or challenges, especially in real-world scenarios, might add depth to the research.

**Questions:**

Would a real-world setting disrupt the assumptions required for analysis and negatively impact performance?

---

> ### Author Response · Authors · 2023-11-23
>
> We are grateful for your valuable feedback on our paper and here are our responses to your comments below.
>
> >W1: The simulations were conducted on the platform: GridWorld. Expanding simulations to more complex or diverse environments can provide a more comprehensive understanding of the algorithm's performance in practice;
>
> A1: Based on your comment, in the revised paper, we have added simulation results in the setting of MountainCar-v1 from OpenAI Gym in the supplementary material. For the mountain car experiment, we first discretize the continuous state spaces with a uniform grid by tile coding. Each tiling has $10 \times 10$ grids. The action can be selected from {push left, no push, and push right}. The simulation results show that the running error is slightly larger than that of I.I.D. sampling in Gridworld with same heterogeneity level ${\kappa ^2} = 1$ . This is because in this setting, there exists discretization error. However, other results coincide with those in Gridworld.
>
>
> >W2: While the authors verified theoretical results in a small-scale problem in a tabular form, it might be valuable to test on larger-scale problems or non-tabular formulations.
>
>
> A2: In fact, the simulations were conducted using linear approximation of value functions. Hence, we have verified the results on non-tabular formulations.
>
> The small-scale testing is conducted due to a better performance guarantee, which helps illustrate the effectiveness of the algorithm. In the case of large-scale testing, however, due to the vast state space, the performance can be more affected by function approximations and designated policies. Then, a more comprehensive and intricate comparison is needed to demonstrate the effectiveness of the algorithm under such scenarios, and we will consider this issue in the future.
>
> >W3: A deeper exploration of the algorithm's limitations or challenges, especially in real-world scenarios, might add depth to the research.
>
> A3: The assumptions used in this paper are common in existing theoretical studies on RL (including FRL). Indeed, some of these assumptions would not hold in practice for real-world applications. For example, linear function approximation (also a common assumption in many existing works) assumes that the true value function can be well approximated by a linear function, using a fixed feature vector $\phi (s)$  for state $s$. However, in real-world settings, it is hard to find a good approximate feature matrix for all states. Also, in this paper, we assume that each agent participates in FRL in each round in a synchronous manner. However, in real-world settings, it can be more efficient for agents to participate in some but not all rounds of FRL in an asynchronous manner. For future work, we will explore the problem of this paper in more general settings by relaxing some of the assumptions used in this paper, such as using non-linear function approximation, and considering partial and asynchronous participation of agents.
>
> >Q1: Would a real-world setting disrupt the assumptions required for analysis and negatively impact performance?
>
> A4: Please refer to A3.

---

> ### Author Response · Authors · 2023-11-23
>
> Dear reviewer: We have added responses to your comments. Could you take a look and let us know if you have any further comment? Thanks!

---

### Official Review · Reviewer_6rEp · 2023-10-31

**Soundness:** 1 poor
**Presentation:** 2 fair
**Contribution:** 1 poor
**Rating:** 3
**Confidence:** 5

**Summary:**

The paper provided a Heterogeneous Federated TD algorithm for finding the optimal value function model for the mixture environments. They presented a sample complexity for both the I.I.D and Markovian sampling.

**Strengths:**

This paper provides a clear and comprehensive investigation into the ongoing challenges of enhancing the sample efficiency of RL agents using federated learning methods in diverse environments.

**Weaknesses:**

The paper has a lot of flaws in analysis.

-  In the abstract, the authors assert that the HFTD achieves linear speedup. Yet, the sample complexity is expressed as $O\left(\frac{1}{\epsilon} \log \frac{1}{\epsilon}\right)$, which doesn't scale with the number of agents $N$. This reveals an inconsistency between the theoretical analysis and the assertions made in the main text.

- The average mean gradient $\bar{g}(\theta)$ doesn't equal to the gradient of Eq(6). This discrepancy arises because $\bar{A}$ is nonsymmetric, whereas a true gradient should be such that the hessian is symmetric. Although Eq. (6) defines the objective of the HFTD algorithm as minimizing the MSPBE, the algorithm actually converges to $\theta^*$, which satisfies the average mean gradient $\bar{g}\left(\theta^*\right)=0$. Hence, $\theta^*$ does not denote the MSPBE's minimum.  As such, HFTD failed in finding the optimal value of the MSPBE.

- The paper's main body contains multiple conflicting statements. The abstract mentions a mixed environment as the average of $N$
heterogeneous environments. However, in Sec 4, it's conveyed that this environment is randomly drawn from the heterogeneous environments. These descriptions are contradictory.

- The authors didn't provide a fair comparison between the results of their paper and those of existing works. In table (1), the objective of Wang (2023) was to find the optimal value function of $i$-th agent, which more focused on the personalization. However, in this paper, the objective function is to find the optimal value function of all $N$ environments, which was in the average sense. If the authors changed the optimality to $i$-th agent's optimal value function,  their methods still cannot converge exactly without the gradient heterogeneity in Assumption 2.

- The bound was quite loose and coarse. The standard convergence result in FL supervised learning [1] is $O(\frac{1}{NTK})$. However, this paper only gave a result of $O(\frac{1}{T}})$, which can not be scaled by $N$ and $K$.

[1] Karimireddy, Sai Praneeth, Satyen Kale, Mehryar Mohri, Sashank Reddi, Sebastian Stich, and Ananda Theertha Suresh. "Scaffold: Stochastic controlled averaging for federated learning." In International conference on machine learning, pp. 5132-5143. PMLR, 2020.

**Questions:**

* What is the dependence on the conditional number $\lambda$ in your sample complexity results? Does this match the existing results in the centralized TD setting [2]?

* What is the dependence on the mixing time for the Markovian sampling in your sample complexity results? Does this match the existing results in the centralized TD setting [2]?

* As mentioned before, $\theta^*$ satisfying $\bar{g}\left(\theta^*\right)=0$ can only find the optimal value function corresponding to the mixture environment, which was a weighted average of $N$ heterogenous environments in Lemma 3. Why should we consider to find this $\theta^*$? What is the motivation of finding $\theta^*$? Because  $\theta^*$ may not equal to the average value function across all agents.

* What is the motivation of doing federation? From the results, the sample complexity can not be scaled by $N$? What's the incentive and benefit for each agent to join the federation?

[2] Bhandari, Jalaj, Daniel Russo, and Raghav Singal. "A finite time analysis of temporal difference learning with linear function approximation." In Conference on learning theory, pp. 1691-1692. PMLR, 2018.

---

> ### Author Response · Authors · 2023-11-23
>
> Thanks for your valuable time and effort in providing such detailed feedback on our work. We hope our responses along with the revised paper can fully address your concerns.
>
> >W1: In the abstract, the authors assert that the HFTD achieves linear speedup. Yet, the sample complexity is expressed as, which doesn't scale with the number of agents $N$. This reveals that there is an inconsistency between the theoretical analysis and the assertions made in the main text.
>
> A1: Yes, the theoretical results were not presented in a way to support the assertions. We are sorry for this confusion. To make the theoretical results consistent with the assertions in the main text, we have revised the bound on the convergence error as an explicit function of $N$, $K$, and $T$ as follows:
> \begin{align}
> {\mathbb{E}\left\| {{{\bar \theta }^T} - {\theta ^*}} \right\|_2^2 \le {\cal{O}}\left( {{e^{ - \sqrt {NKT} }}} \right) + {\cal{O}}\left( {\frac{{N(K - 1)}}{T}} \right) + {\cal{O}} \left( {\frac{1}{{\sqrt {NKT} }}} \right)}.
> \end{align}
>
> When the number of communication rounds $T$ is sufficiently large, the convergence error is dominated by the last term ${\cal{O}}({\frac{1}{{\sqrt {NKT} }}})$. Then we can conclude that the total sample complexity for achieving an $\epsilon$-accurate optimal solution ${\mathbb{E}}\left\| {{{\bar \theta }^T} - {\theta ^*}} \right\|_2^2 \le \varepsilon $ is $KT = {\cal{O}}\left( {\frac{1}{{N{\varepsilon ^2}}}} \right)$. Therefore, the sample complexity scales with the number of agents $N$.
>
> >W2: The average mean gradient $\bar g(\theta )$ doesn't equal to the gradient of Eq(6) as $\mathop {\min }\limits_{\theta  \in {{\mathbb{R}}^d}} \left[ {F(\theta ) = \frac{1}{N}\sum\limits_{i = 1}^N {{F_i}(\theta )} } \right]$. This discrepancy arises because $\overline A $ is nonsymmetric, whereas a true gradient should be such that the hessian is symmetric. Although Eq.(6) defines the objective of the HFTD algorithm as minimizing the MSPBE, the algorithm actually converges to ${\theta ^*}$, which satisfies the average mean gradient $\bar g({\theta ^*}) = 0$. Hence, HFTD failed in finding the optimal value of the MSPBE.
>
> A2: For each agent $i$, the function ${{\bar g_i}}(\theta )$ is similar to a "gradient" but is actually *not* the true gradient of any fixed objective function. This is because the "gradient descent" step, $g_i(\theta_t ) = (r_t + \gamma \phi {({s_{t + 1}})^{\rm{T}}}\theta  - \phi {(s_t)^{\rm{T}}}\theta )\phi (s_t)$, taken at time $t$ makes the estimated value ${V_\theta^i }(s_t)$ closer to the target $r_t^i + \gamma {V_\theta^i }({s_{t + 1}})$, but $r_t^i + \gamma {V_\theta^i }({s_{t + 1}})$ itself depends on $V_\theta^i $. When there is no confusion, we omit the subscript $i$ in the discussion below. It has been proven in [3] that the projected Bellman operator is a contraction operator as ${\left\| {{\Pi_D}{T_\mu }{V_{\theta_1} }} - {{\Pi_D}{T_\mu }{V_{\theta_1} }} \right\|_D}$ $\le$
>
> $\gamma {\left\| {{V_{\theta_1} }-{V_{\theta_2} }} \right\|_D}$.
>
> By repeatedly applying the contraction, one can obtain the approximate optimal value function $\Phi {\theta ^*}$ as
>
> ${\left\| {{V_{{\theta ^*}}} - {V_\mu }} \right\|_D}$ $\le$
>
>  $\frac{1}{{\sqrt {1 - {\gamma ^2}} }}{\left\| {{\Pi_D}{V_\mu } - {V_\mu }} \right\|_D}$
>
> It measures the root-mean squared deviation between the value predictions of the limiting TD value function and the true value function ${{V_\theta }}$. This means that, although ${{\bar g_i}}(\theta )$ is not the true gradient of the MSPBE ${F_i}(\theta ) $,
> the value $\theta _i^*$ that satisfies  $\bar g_i(\theta _i^*) = 0$ is still the optimal solution that minimizes $F_i(\theta)$. Moreover, a powerful ODE method was provided in [3] using the stochastic approximation $\dot \theta  = {{\bar g}_i}(\theta ) = {{\bar A}_i}\theta  + {{\bar b}_i}$ where ${{\bar A}_i}$ is negative definite and symmetric. Thus, this ODE is exponentially stable and $\theta$ will converge to a globally and asymptotically stable equilibrium point ${\theta_i ^*}$ where $\theta _i^* =  - \bar A_i^{ - 1}{{\bar b}_i}$. Therefore, although $\bar g(\theta )$ is not the true gradient of (6), $\bar g({\theta ^*}) = 0$ is still the optimal solution to (6) (Eq.(5) in the revised paper).

---

> ### Author Response · Authors · 2023-11-23
>
> >W3: The paper's main body contains multiple conflicting statements. The abstract mentions a mixed environment as the average of $N$ heterogeneous environments. However, in Section 4, it's conveyed that this environment is randomly drawn from the heterogeneous environments. These descriptions are contradictory.
>
> A3: Based on your comment, in the revised paper, we have revised the abstract to clarify that the "mixture of environments" is the environment *randomly drawn* from agents' heterogeneous environments (marked in red in the revised paper). This *mixture* environment is different from the *single virtual* environment $\( S,A,\bar P,\bar R,\gamma\)$ defined in [5-6], which is obtained from averaging the transition kernels and reward functions of agents' environment as $\overline P  = \frac{1}{N}\sum\nolimits_{i = 1}^N {{P_i}} $ and $\bar R = \frac{1}{N}\sum\nolimits_{i = 1}^N {{R_i}} $. The goal of this paper is to find the optimal value function model for the mixture environment of agents. This goal is similar in spirit to that of federated supervised learning, as the latter aims to find the optimal model that minimizes the average training loss for all data samples of all clients.
>
> >W4: The authors didn't provide a fair comparison between the results of their paper and those of existing works. In table (1), the objective of Wang (2023) [6] was to find the optimal value function of $i$-th agent, which more focused on the personalization. However, in this paper, the objective function is to find the optimal value function of all $N$ environments, which was in the average sense. If the authors changed the optimality to $i$-th agent's optimal value function, their methods still can not converge exactly without the gradient heterogeneity in Assumption 2.
>
> A4: We compare the results of this paper with those of related existing works as follows. The goal of Wang (2023) was to find the optimal value function for the *single virtual* environment $\( S,A,\bar P,\bar R,\gamma\)$  where $\overline P  = \frac{1}{N}\sum\nolimits_{i = 1}^N {{P_i}} $ and $\bar R = \frac{1}{N}\sum\nolimits_{i = 1}^N {{R_i}} $. They obtained the personalization result $\mathbb{E}\left\| {{V_{\theta_T} } - {V_{{\theta_i} ^*}}} \right\|_{\bar D}^2$. It is bounded by $T_1$ and $T_2$, where $T_1$ denotes
>
> $\mathbb{E}\left\| {{V_{\theta_T} } - {V_{\theta ^*}}} \right\|_{\bar D}^2$, and $T_2$ denotes
>
> $\mathbb{E}\left\| {{V_{{\theta_i} ^*} } - {V_{{\theta ^*}}}} \right\|_{\bar D}^2$,
>
>
> where $\bar D$ is the stationary distribution of the averaged MDP $\( S,A,\bar P,\bar R,\gamma\)$; ${\theta ^*}$ is the optimal solution to the value evaluation problem of the averaged MDP. Nevertheless, $T_1$ and $T_2$ both involve some non-vanishing terms. Specifically, [6] focused on the term $T_1$. In handling per round progress by using the update rule, the inner product includes $\left\| {\bar g(\theta ) - \frac{1}{N}\sum\limits_{i = 1}^N {{{\bar g}_i}(\theta )} } \right\|$ where ${\bar g({{\bar \theta^t }})}$ is the expected gradient of the *virtual* MDP. It is non-zero, seeing Lemma 2 in [6]. That is the reason why $T_1$ has non-vanishing terms. For $T_2$, it is non-vanishing due to environment heterogeneity as $\theta _i^* \ne {\theta ^*}$.
>
> Compared to [6], in this paper, we consider the \textit{mixture} environment which is the environment \textit{randomly drawn} from agents' heterogeneous environments, rather than the single virtual environment [6] mentioned above. Based on this setting, we propose the HFTD algorithm which aims to iteratively find the optimal value function model that minimizes the average MSPBE of agents' heterogeneous environments. Therefore, the objective function fo the FRL problem in this paper is different from that in [6]. We show that the HFTD algorithm can asymptotically achieve the optimal model in the settings of this paper. We do not claim that the HFTD algorithm can asymptotically achieve the optimal model for the learning problem in [6]. Also, the HFTD algorithm would not asymptotically achieve the local optimal model for any agent due to environment heterogeneity.
>
> >W5: The bound was quite loose and coarse. The standard convergence result in FL supervised learning [1] is $O\left( {\frac{1}{{NTK}}} \right)$, However, this paper only gave a result of $O\left( {\frac{1}{T}} \right)$, which can not be scaled by $N$ and $K$.
> .
>
> A5: Yes, the bound on the convergence error was not presented in a way to support the claims. We are sorry for this confusion. Please refer to the response A1 for detailed discussions.

---

> ### Author Response · Authors · 2023-11-23
>
> >Q1: What is the dependence on the conditional number $\lambda$ in your sample complexity results? Does this match the existing results in the centralized TD setting [2]?
>
> A6: In this paper, when agents have a constant local update number $K$, the convergence result is:
> \begin{align*}
> \mathbb{E}\left\| {{{\bar \theta }^T} - {\theta ^*}} \right\|_2^2 \le {e^{ - \frac{{\lambda \alpha KT}}{8}}}\left\| {{{\bar \theta }^0} - {\theta ^*}} \right\|_2^2 + \frac{{256{\alpha ^2}{\sigma ^2}(K - 1)}}{{{\lambda ^2}}} + \frac{{512{\alpha ^2}{\kappa ^2}K(K - 1)}}{{{\lambda ^2}}} + \frac{{16\alpha {\sigma ^2}}}{{\lambda N}}
> \end{align*}
>
> In the centralized TD setting which corresponds to $N=1$ and $K=1$ in the settings of this paper, the sample complexity of the HFTD algorithm in this paper is $O\left( {\frac{1}{{{\lambda ^2}{\varepsilon ^2}}}} \right)$. [2] did not provide the result of the sample complexity. From Theorem 2(a)(b) in [2], we can obtain that the sample complexity is $O\left( {\frac{1}{{{{(1 - \gamma )}^2}{\varepsilon ^2}}}} \right)$. In our paper, we have $\lambda = 1-\gamma$ which follows from Lemma 3 and Lemma 4 in [2]. Hence, the sample complexity result in this paper matches the existing results in the centralized TD setting [2]. We have added the comparison discussed above in Remark 4 in the revised paper (marked in red).
>
> >Q2: What is the dependence on the mixing time for the Markovian sampling in your sample complexity results? Does this match the existing results in the centralized TD setting [2]?
>
> A7: In the centralized setting which corresponds to $N=1$ and $K=1$ in the settings of this paper, the convergence error bound of the HFTD algorithm in this paper is
> \begin{align}
> \mathbb{E}\left\| {{{\bar \theta }^T} - {\theta ^*}} \right\|_2^2 \le {e^{ - \frac{{\lambda \alpha T}}{4}}}\left\| {{{\bar \theta }^0} - {\theta ^*}} \right\|_2^2 + \frac{{2\alpha H\left( {2c'H + q' + 4{\beta ^2}{\tau ^2}cH} \right)}}{\lambda }+\frac{{8\alpha \left( {3 + 2{\tau ^2} } \right){{[q']}^2}}}{\lambda }
> \end{align}
> which depends on ${{\tau ^2}}$. [2] did not provide the result of the sample complexity. The convergence error bound in [2] is:
> \begin{align}
> \mathbb{E}\left\| {{\theta ^T} - {\theta ^*}} \right\|_2^2 \le \left( {{e^{ - {\alpha _0}(1 - \gamma )\omega T}}} \right)\left\| {{\theta ^T} - {\theta ^0}} \right\|_2^2 + {\alpha _0}\left( {\frac{{{G^2}(9 + 12{\tau ^{mix}}({\alpha _0}))}}{{2(1 - \gamma )\omega }}} \right)
> \end{align}
>
>
> which depends on ${{\tau}}$. Compared to the result in [2], the difference of the result of this paper is because we need to decompose ${\mathbb{E}_{t - 2\tau }}\left\| {{\theta ^t} - {\theta ^{t - \tau }}} \right\|_2^2$ to obtain ${\alpha ^2}$, so that the bound does not include any non-vanishing term.
>
> >Q3: As mentioned before, ${\theta ^*}$  satisfying $g({\theta ^*}) = 0$ can only find the optimal value function corresponding to the mixture environment, which was a weighted average of $N$ environments in Lemma 3. Why should we consider to find this ${\theta ^*}$? What is the motivation of finding ${\theta ^*}$? Because ${\theta ^*}$ may not equal to the average value function across all agents.
>
> A8: [5-6] both considered an averaged environment which averages the transition probability kernel and reward function. However, in Lemma 3, we have proved that there exists a Markov chain whose expected gradient calculated by temporal difference learning can be expressed by Eq(6). Note that this Markov chain does not correspond to the mixed environment by averaging $N$ agents' environments [5-6]. In this paper, since ${\theta ^*} \ne \frac{1}{N}\sum\limits_{i = 1}^N {\theta _i^*}$ due to heterogeneous environments, ${\theta ^*}$ may not correspond to the average value function across all agents. However, ${\theta ^*}$ is the value evaluation model which can minimize the averaged MSPBE across the $N$ environments. That is to say, ${\theta ^*}$ can minimize the expected TD error with respect to the environment randomly drawn from the $N$ environments.

---

> > ### Author Response · Authors · 2023-11-23
> >
> > >Q4: What is the motivation of doing federation? From the results, the sample complexity cannot be scaled by $N$? What's the incentive and benefit for each agent to join the federation?
> >
> > A9: The dependency on $N$ was not clearly shown in the variance reduction term. We have revised it. Now the sample complexity can be scaled by $N$. Please refer to A1.
> >
> > In RL, an agent needs to learn the optimal policy in an online manner from a large amount of data samples via repeatedly interacting with the environment to achieve a high learning accuracy and it suffers from computational inefficiency. With federation, agents can accelerate the process of learning based on other's experience.
> >
> > This paper aims to find the optimal value function model for the mixture environment when the environment is randomly drawn from the heterogeneous environments. The results show that the sample complexity of each agent achieves linear speedup with $N$ using federation. Although the proposed algorithm would not asymptotically achieve the local optimal model, it produces an optimal model for the aggregated environment. This can also be further utilized for local model personalization.
> >
> >
> > >References
> >
> > [1] Karimireddy, Sai Praneeth, et al. "Scaffold: Stochastic controlled averaging for federated learning." in ICML, 2020.
> >
> > [2] Bhandari, Jalaj, Daniel Russo, and Raghav Singal. "A finite time analysis of temporal difference learning with linear function approximation." In CLOT, 2018.
> >
> > [3] Tsitsiklis, John N., and Benjamin Van Roy. "An Analysis of Temporal-Difference Learning with Function Approximation." IEEE Transactions On Automatic Control 42.5, 1997.
> >
> > [4] Sun, Jun, et al. "Finite-time analysis of decentralized temporal-difference learning with linear function approximation." in AISTATS, 2020
> >
> > [5] Jin H, Peng Y, Yang W, et al. "Federated reinforcement learning with environment heterogeneity." in AISTATS, 2022
> >
> > [6] Wang, Han, et al. "Federated temporal difference learning with linear function approximation under environmental heterogeneity." arXiv preprint arXiv:2302.02212, 2023.

---

> ### Author Response · Authors · 2023-11-23
>
> Dear reviewer: We have added responses to your comments. Could you take a look and let us know if you have any further comment? Thanks!

---

### Official Review · Reviewer_X2Ca · 2023-11-04

**Soundness:** 2 fair
**Presentation:** 1 poor
**Contribution:** 2 fair
**Rating:** 3
**Confidence:** 4

**Summary:**

This paper studies the federated RL problem, and designs a HFTD algorithm for federated TD learning with linear function approximation under environment heterogeneity and computation heterogeneity. It is shown that the HFTD algorithm can asymptotically converge to the optimal value function model achieving linear speedup in convergence. Numerical validations are provided.

**Strengths:**

+ A federated RL setting with heterogeneous environments and computations.
+ A heterogeneous federated TD learning algorithm.
+ Non-asymptotic convergence analysis.

**Weaknesses:**

- The presentation of the paper can be improved with a lot of grammar issues and inaccurate statements (see Questions below for details).  For example, there shall be citations for the statement "Hence, inspired by FL, federated reinforcement learning (FRL) has been proposed as a promising approach which allows agents to collectively learn the policy without sharing raw data samples." In the paper, only one citation on federated RL is provided (Wang et al. arXiv 2023). It is difficult to claim that FRL has been a "promising" approach. Overall, I am not convinced by merging the FL setup with RL. If every agent faces heterogeneous MDPs, what is the goal of the FRL?
- The algorithm is a combination of the FL and TD learning algorithm; and the analysis also follows mostly from existing literature.

**Questions:**

1) There are quite a lot typos/grammar issues; See e.g., "each agents in FRL collects" "none of these works have shown" “design a federated TD algorithm that asymptotically converge to” “how much is the sample complexity of this algorithm? ” "We provide numeral results"
2) The paper claims that using federated multiple agents to collaboratively evaluate the policy can accelerate the convergence achieving linear speedup. It would be great if numerical results can be provided to demonstrate this linear speedup under heterogenous settings.

---

> ### Author Response · Authors · 2023-11-23
>
> We are grateful for your valuable comments on our paper. We have carefully prepared responses to your comments as below. We hope they can fully address your concerns.
>
> **W1: The presentation of the paper can be improved with a lot of grammar issues and inaccurate statements. For example, there shall be citations for the statement "Hence, inspired by FL, federated reinforcement learning (FRL) has been proposed as a promising approach which allows agents to collectively learn the policy without sharing raw data samples." In the paper, only one citation on federated RL is provided (Wang et al. arXiv 2023). It is difficult to claim that FRL has been a "promising" approach. Overall, I am not convinced by merging the FL setup with RL. If every agent faces heterogeneous MDPs, what is the goal of the FRL?**
>
> A1: RL faces severe sample inefficiency in real systems. For example, when RL is applied to train a walking robot, the walking ability of the robot is limited by the number of samples used by the robot. A natural idea to deal with the sample-efficiency issue of RL is federated RL (FRL) which allows agents to share the information so that agents can collectively learn a better policy. This policy may perform slightly worse than the optimal policy for each local agent, but it can possess a broad applicability to effectively address all local objectives. With federation, agents can accelerate the process of learning based on other's experience.
>
> We have presented thorough investigation on FRL with theoretical guarantees [1-7] in the part of federated learning as an independent subsection in Related Works. Besides, we have added citations for the statement "Hence, inspired by FL, federated reinforcement learning (FRL) has been proposed as a promising approach which allows agents to collectively learn the policy without sharing raw data samples, to address the sample inefficiency of RL in real systems" [1-7]. We marked the revisions in red.
>
> **W2: The algorithm is a combination of the FL and TD learning algorithm; and the analysis also follows mostly from existing literature.**
>
> A2: The HFTD algorithm proposed in this paper is much more than a simple combination of FL and TD learning. There are three major differences compared to the existing works on FRL [1-7], discussed as follows.
>
> 1) The FRL settings considered in this paper have non-trivial differences from those in prior works on FRL. Compared to parallel RL where agents interact with identical environments [1, 2, 6], FRL where agents have heterogeneous environments remains largely under-explored. In a few recent works on FRL [3-4] for agents with heterogeneous environments, the objective is to find the optimal value function model for a *single virtual* environment $\( S,A,\bar P,\bar R,\gamma\)$ obtained from averaging the transition kernels and reward functions of agents' environment as $\overline P  = \frac{1}{N}\sum\nolimits_{i = 1}^N {{P_i}} $ and $\bar R = \frac{1}{N}\sum\nolimits_{i = 1}^N {{R_i}} $. However, such an ``averaged'' virtual environment may not coincide with the individual environment of any of the agents for FRL. In this paper, we consider a *mixture* environment defined as the environment *randomly drawn* from agents’ heterogeneous environments. Note that this mixture environment is different from the virtual environment [3-4] mentioned above. Thus, our goal is to find the optimal value function model for this mixture environment of agents. This goal is similar in spirit to that of federated supervised learning, as the latter aims to find the optimal model that minimizes the average training loss for all data samples of all clients.
>
> 2) The HFTD algorithm can asymptotically achieve the optimal value function model in the FRL settings considered in this paper. In existing works on FRL for heterogeneous environments [3-4], the bound of the convergence error is non-vanishing. In this paper, to our best knowledge, we provide the first result in existing works on FRL with heterogeneous environments that the convergence error can diminish to zero asymptotically. This result is because that the goal of the HFTD algorithm is to find the optimal value function model for the mixture environment that is randomly drawn from agents’ heterogeneous environments. To this end, the algorithm aims to minimize the average of agents’ mean squared projected Bellman errors (MSPBEs) for their respective environments. In particular, a key property of the global gradient of the average MSPBE allows us to remove a non-vanishing bias in the convergence error, so that only vanishing error terms remain. We also show that the HFTD algorithm achieves a sample complexity of ${\cal{O}}\left( {\frac{1}{{N{\varepsilon ^2}}}} \right)$ and a linear convergence speedup, which match the results of existing TD learning algorithms [6, 8].

---

> ### Author Response · Authors · 2023-11-23
>
> 3) The HFTD algorithm allows agents to use different numbers of local iterations based on their capabilities of collecting samples. The existing works on FRL [1-4] assume that agents use the same number of local iterations in each round of the algorithm. However, in practice, agent's capabilities of sample collection and local computation can be highly diverse, so that stragglers can significantly slow down the training process. Therefore, it is more efficient for agents to use different numbers of local iterations based on their respective capabilities.
> ---
> **Q1: There are quite a lot of typos/grammar issues.**
>
> A3: In the revised paper, we have corrected all grammatical errors and typos (marked in red in the revised paper). For example, "each agents in FRL collects" is changed to "each agent in FRL collects", "none of these works have shown" is changed to "none of these works has shown", "design a federated TD algorithm that asymptotically converge to" is changed to "design a federated TD algorithm that asymptotically converges to", "we provide numeral results" is changed to "we provide numerical results".
>
> **Q2: It would be great if numerical results can be provided to demonstrate this linear speedup under heterogeneous settings.**
>
> A4: Indeed, these numerical results were all conducted under the heterogeneous setting as Fig 1. Furthermore, Fig 1(c) demonstrates this linear speedup under heterogeneous settings. Following your suggestion, we have added more details of heterogeneous settings. Assumption 2 defines the environment heterogeneity shown as bounded gradient heterogeneity, $\frac{1}{N}\sum\nolimits_i {\left\| {{{\bar g}_i}(\theta )} \right\|_2^2}  \le {\beta ^2}\left\| {\frac{1}{N}\sum\nolimits_i {{{\bar g}_i}(\theta )} } \right\|_2^2 + {\kappa ^2}$ where ${\kappa ^2}$ denotes the heterogeneity level. Here we conduct the simulations with ${\kappa ^2}=1$. Based on the fact that ${{\bar g}_i}(\theta ) = {\Phi ^{\rm{T}}}{D_i}(\gamma {P_i}\Phi  - \Phi )\theta  + {\Phi ^{\rm{T}}}{D_i}{R_i}$, a series of heterogeneous transition probability kernel and reward function are generated $\( {P_i},{R_i}\) _{i = 1}^N$ to satisfy Assumption 2. After that, these environments are executed on GridWorld. Fig 1(c) shows that one achieves a linear speedup with respect to the number of workers which matches our theoretical result ${\mathbb{E}\left\| {{{\bar \theta }^T} - {\theta ^*}} \right\|_2^2 \le {\cal{O}}\left( {{e^{ - \sqrt {NKT} }}} \right) + {\cal{O}}\left( {\frac{{N(K - 1)}}{T}} \right) + {\cal{O}} \left( {\frac{1}{{\sqrt {NKT} }}} \right)}$. Typically, when $T$ is sufficiently large, the total sample complexity for achieving an $\epsilon$-accurate optimal solution ${\mathbb{E}}\left\| {{{\bar \theta }^T} - {\theta ^*}} \right\|_2^2 \le \varepsilon $ is $KT = {\cal{O}}\left( {\frac{1}{{N{\varepsilon ^2}}}} \right)$.
>
>
>
> **References**
>
> [1] Fan, Xiaofeng, et al. "Fault-tolerant federated reinforcement learning with theoretical guarantee." in NeuIPS'21
>
> [2] Khodadadian S, Sharma P, Joshi G, et al. "Federated reinforcement learning: Linear speedup under markovian sampling" in ICML'22
>
> [3] Jin H, Peng Y, Yang W, et al. "Federated reinforcement learning with environment heterogeneity." in AISTATS'22
>
> [4] Wang, Han, et al. "Federated temporal difference learning with linear function approximation under environmental heterogeneity." arXiv preprint arXiv:2302.02212, 2023.
>
> [5] Zeng, Sihan, et al. "A decentralized policy gradient approach to multi-task reinforcement learning." in UAI'21.
>
> [6] Sun, Jun, et al. "Finite-time analysis of decentralized temporal-difference learning with linear function approximation." in AISTATS'20.
>
> [7] Yang, Tong, et al. "Federated Natural Policy Gradient Methods for Multi-task Reinforcement Learning." arXiv preprint arXiv:2311.00201, 2023.
>
> [8] Bhandari, Jalaj, Daniel Russo, and Raghav Singal. "A finite time analysis of temporal difference learning with linear function approximation." In CLOT, 2018.

---

> ### Author Response · Authors · 2023-11-23
>
> Dear reviewer: We have added responses to your comments. Could you take a look and let us know if you have any further comment? Thanks!

---

### Meta-Review · Area_Chair_hyBq · 2023-12-19

**Metareview:**

The paper studies Temporal Difference (TD) learning with linear function approximation in a federated setting where multiple agents collaboratively perform policy evaluation via TD learning while interacting with heterogeneous environments and using heterogeneous computation configurations.

I thank the authors for their thorough responses. The reviewers had raised several concerns, some alleviated by the responses, but some others remained. Furthermore, after the discussion period, some of the reviewers felt that the paper would need quite some work to incorporate all the comments and hence it may be better to prepare a new version for future conferences.  I encourage the authors to incorporate the comments (e.g. on the correctness of the statements, comparisons, etc--see the updated reviews and comments by the reviewers) from the reviewers. Once the reviewers' comments are fully addressed, the paper will be an excellent contribution to the FL+RL communities.

**Justification For Why Not Higher Score:**

I recommended rejecting -- the decision is based on the reviews and the discussions afterwards.

**Justification For Why Not Lower Score:**

--

---

### Decision · Program_Chairs · 2024-01-16

Reject